


# Energy-conservation datasets of global land surface radiation and heat fluxes from 2000-2020 generated by CoSEB

Junrui Wang[a, b], Ronglin Tang[a, b, *], Meng Liu[c], Zhao-Liang Li[a, b, c]

[a] State Key Laboratory of Resources and Environment Information System, Institute of Geographic Sciences and Natural Resources Research, Chinese Academy of Sciences, Beijing 100101, China

[b] University of Chinese Academy of Sciences, Beijing 100049, China

[c] State Key Laboratory of Efficient Utilization of Arable Land in China, Institute of Agricultural Resources and Regional Planning, Chinese Academy of Agricultural Sciences, Beijing 100081, China

* Authors to whom correspondence should be addressed: tangrl@lreis.ac.cn

**Abstract**

Accurately estimating global land surface radiation [including downward shortwave radiation ($SW_{IN}$), downward longwave radiation ($LW_{IN}$), upward shortwave radiation ($SW_{OUT}$), upward longwave radiation ($LW_{OUT}$) and net radiation (Rn)] and heat fluxes [including latent heat flux (LE), soil heat flux (G) and sensible heat flux (H)] is essential for quantifying the exchange of radiation, heat and water between the land and atmosphere under global climate change. This study presents the first energy-conservation datasets of global land surface radiation and heat fluxes from 2000 to 2020, generated by our model of Coordinated estimates of land Surface Energy Balance components (CoSEB) that was renewed with a combination of GLASS and MODIS remote sensing data, ERA5-Land reanalysis datasets, topographic data, $CO_2$ concentration data, and observations at 258 eddy covariance sites worldwide from the AmeriFlux, FLUXNET, EuroFlux, OzFlux, ChinaFLUX and TPDC. The developed CoSEB-based datasets are strikingly advantageous in that [1] they are the first RS-based global datasets that satisfy both surface radiation balance ($SWI_N$ - $SW_{OUT}$ + $LW_{IN}$ - $LW_{OUT}$ = Rn) and heat balance (LE + H + G = Rn) among the eight fluxes, as



demonstrated by both the radiation imbalance ratio [RIR, defined as 100% × ($SW_{IN}$ −
$SW_{OUT}$ + $LW_{IN}$ - $LW_{OUT}$)/Rn] and energy imbalance ratio [EIR, defined as 100% × (Rn
- G - LE - H)/Rn] of 0, [2] the radiation and heat fluxes are characterized by high
accuracies, where (1) the RMSEs for daily estimates of $SW_{IN}$, $SW_{OUT}$, $LW_{IN}$, $LW_{OUT}$, Rn,
LE, H and G from the CoSEB-based datasets were 28.51 W/m$^2$, 10.39 W/m$^2$, 14.29
W/m$^2$, 10.62 W/m$^2$, 22.40 W/m$^2$, 24.38 W/m$^2$, 22.67 W/m$^2$ and 6.77 W/m$^2$, respectively,
as well as for 8-day estimates were 12.81 W/m$^2$, 7.08 W/m$^2$, 9.22 W/m$^2$, 8.34 W/m$^2$,
13.38 W/m$^2$, 19.99 W/m$^2$, 17.44 W/m$^2$ and 4.25 W/m$^2$, respectively, (2) the CoSEB-
based datasets, in comparison to the mainstream products/datasets (i.e. GLASS, BESS-
Rad, BESSV2.0, FLUXCOM, MOD16A2, PML_V2 and ETMonitor) that generally
separately estimated subsets of the eight flux components, better agreed with the in situ
observations. Our developed datasets hold significant potential for application across
diverse fields such as agriculture, forestry, hydrology, meteorology, ecology, and
environmental science, which can facilitate comprehensive studies on the variability,
impacts, responses, adaptation strategies, and mitigation measures of global and
regional land surface radiation and heat fluxes under the influences of climate change
and human activities. The CoSEB-based datasets are open access and available through
the National Tibetan Plateau Data Center (TPDC) at
https://doi.org/10.11888/Terre.tpdc.302559 (Tang et al., 2025a) and through the Science
Data Bank (ScienceDB) at https://doi.org/10.57760/sciencedb.27228 (Tang et al.,
2025b).
**Key words:** Surface energy balance; Surface radiation balance; Shortwave/Longwave
radiation; Net radiation; Sensible/Latent heat flux; Evapotranspiration; CoSEB
**1 Introduction**

Land surface radiation balance and heat balance play important roles in Earth's

climate system, representing the physical processes by which the surface-atmosphere
absorbs and redistributes radiation and heat fluxes (Berbery et al., 1999; Betts et al.,
1996; Mueller et al., 2009; Sellers et al., 1997; Xu et al., 2022a), and facilitating the



exchange of water, energy, carbon, and other agents essential to climatic and ecological
systems and human society (Jia et al., 2013; Wang et al., 2012; Wild, 2009; Wild et al.,
2012; Xia et al., 2006). Accurately monitoring the spatial and temporal variations of
global land surface radiation [including downward shortwave radiation ($SW_{IN}$),
downward longwave radiation ($LW_{IN}$), upward shortwave radiation ($SW_{OUT}$), upward
longwave radiation ($LW_{OUT}$) and net radiation (Rn)] and heat fluxes [including latent
heat flux (LE), soil heat flux (G) and sensible heat flux (H)] is indispensable for
quantifying the exchange of radiation, heat and water between the land and atmosphere
under global climate change (Ersi et al., 2024; Liang et al., 2019; Rios & Ramamurthy,
2022; Tang et al., 2024a; Wang et al., 2021), and for studying solar energy utilization
(Tang et al., 2024b; Zhang et al., 2017), hydrological cycle (Huang et al., 2015; Wild &
Liepert, 2010), ecosystem productivity (Nemani et al., 2003), agricultural management
(De Wit et al., 2005) and ecological protection (Tang et al., 2023). Remote sensing (RS)
technology, with its high spatial-temporal resolution and applicability over large areas,
is considered to be the most effective and economical means for obtaining global land
surface radiation and heat fluxes (Liu et al., 2016; Van Der Tol, 2012; Zhang et al.,

2010).

In past decades, numerous RS-based products/datasets of global surface radiation

and heat fluxes have significantly advanced, which were generally generated by
physical or statistical methods (Jiao et al., 2023; Jung et al., 2019; Martens et al., 2017;
Yu et al., 2022). However, two key limitations still exist in these products. Firstly, most
available products provide only a single component of land surface radiation or heat
fluxes, e.g. ETMonitor (Zheng et al., 2022) and MOD16A2 (Mu et al., 2011) only
estimating LE, leading to the failure to satisfy surface radiation balance and heat
balance when the single radiation or heat flux is utilized in conjunction with products
containing other radiation and heat components (Wang et al., 2025), and further posing
significant uncertainties to understand the interactions and redistributions of surface
radiation and energy in the Earth-atmosphere system. Secondly, a few products, e.g.



FLUXCOM (Jung et al., 2019) and GLASS (Jiang et al., 2015; Zhang et al., 2014),
generated datasets for multiple components of surface radiation and heat fluxes by using
independent estimates from the uncoordinated models, which make them difficult to
abide by surface radiation and heat conservation. These energy-imbalanced and
radiation-imbalanced estimates among multiple components from previous
products/datasets severely limit their in-depth applications in analyzing the spatial and
temporal trends, simulating the physical process of radiation, heat and water cycles as
well as revealing the attributions and mechanisms in Earth-surface system under global
climate change. It was impending and imperative to develop global datasets of land
surface radiation and heat fluxes characterized by high accuracies, radiation balance as
well as heat balance, to better meet the requirements in practical applications of various
fields.

Our proposed data-driven model/framework of Coordinated estimates of land

Surface Energy Balance components (CoSEB) (Wang et al., 2025), which effectively
learns the underlying physical interrelations (i.e. surface energy conservation law)
among multiple targeted variables, provides an unprecedented opportunity to develop
global datasets of land surface radiation and heat fluxes that can not only
simultaneously provide high-accuracy estimates of these components but also adhere
to surface radiation- and heat-conservation laws.

The objectives of this study are twofold: (1) to develop high-accuracy datasets of

global land surface radiation and heat fluxes, which comply with the principles of
radiation balance and heat balance, using our CoSEB model renewed based on in situ
observations, remote sensing data and reanalysis datasets; (2) to validate the
datasets/model estimates against data from in situ observations, mainstream products
as well as estimates from uncoordinated random forest (RF) techniques. Section 2
introduces the data resources used in this study. Section 3 briefly describes the method
we used to estimate global surface radiation and heat fluxes. Section 4 presents the
evaluation of the datasets/model estimates generated by our renewed CoSEB model.





Section 5 discusses the superiority, potential applications and uncertainties of the
developed datasets. Data availability is given in Section 6, and a summary and
conclusion is provided in Section 7.
**2 Data**
**2.1 Ground-based observations**

In this study, the in situ observations of land surface radiation and heat fluxes at

258 eddy covariance (EC) sites from the networks of AmeriFlux (145 sites, 2000–2020,
https://AmeriFlux.lbl.gov/Data/, last access: 6 August 2024), EuroFlux (72 sites, 2000-
2020, http://www.europe-fluxdata.eu/, last access: 6 August 2024), OzFlux (5 sites,
2007-2012, https://data.ozflux.org.au/, last access: 6 August 2024), FLUXNET (108
sites, 2000–2014, https://FLUXNET.org/Data/download-Data/, last access: 6 August
2024), ChinaFLUX (5 sites, 2005-2020, http://www.chinaflux.org/, last access: 6
August 2024) and National Tibetan Plateau/Third Pole Environment Data Center
(TPDC, 13 sites, 2012–2020, https://Data.tpdc.ac.cn/en/Data, last access: 6 August
2024) were used (Fig. 1), where 37, 48 and 5 sites in FLUXNET were also shared in
AmeriFlux, EuroFlux and OzFlux, respectively. These 258 sites were filtered out from
all collected 1008 sites by following the quality-assurance and quality-control steps,
including: (1) any site with a missing component of any of the $SW_{IN}$, $SW_{OUT}$, $LW_{IN}$,
$LW_{OUT}$, LE, H and G was excluded, reducing the 1008 sites to 388 sites for further
analysis; (2) any half-hour period with missing data for any of these components was
excluded; (3) the half-hourly ground-based observations with quality-control flag of 2
or 3 (bad quality) were removed but quality-control flag of 0 and 1 (good quality) were
maintained; (4) a daily average of the half-hour observations was calculated for each
day with greater than 80% good-quality data, further reducing the 388 sites to 286 sites;
(5) the aggregated daily LE and H were corrected for energy imbalance using the
Bowen ratio method when the daily energy balance closure [defined as
$(LE+H)/(Rn-G)$ ] varied between 0.2 and 1.8; (5) outliers were discarded,



corresponding to the 1 and 99 quantiles of the daily evaporation fraction, further
reducing the 286 sites to 268 sites. Besides, the RS data involved in this study collocated
at the sites should not be missing, finally reducing the 268 sites to 258 sites for analysis.
Note that the Rn at these sites used in this study was calculated from the sum of net
longwave radiation ($LW_{IN}$ minus $LW_{OUT}$) and net shortwave radiation ($SW_{IN}$ minus
$SW_{OUT}$), rather than using the observed Rn directly, to ensure surface radiation balance
in training datasets.
These 258 sites used in this study cover a wide range of global climate regimes
across 14 land cover types, including (1) evergreen needleleaf forests (ENF, 54 sites);
(2) evergreen broadleaf forests (EBF, 11 sites); (3) deciduous needleleaf forests (DNF,
1 sites); (4) deciduous broadleaf forests (39 sites); (5) mixed forests (MF, 8 sites); (6)
closed shrublands (CSH, 5 sites); (7) open shrublands (OSH, 9 sites); (8) woody
savannas (WSA, 6 sites); (9) savannas (SAV, 10 sites); (10) grasslands (GRA, 54 sites);
(11) permanent wetlands (WET, 16 sites); (12) croplands (CRO, 43 sites); (13) water
bodies (WAT, 1 sites); (14) cropland/natural vegetation mosaics (CVM, 1 sites).

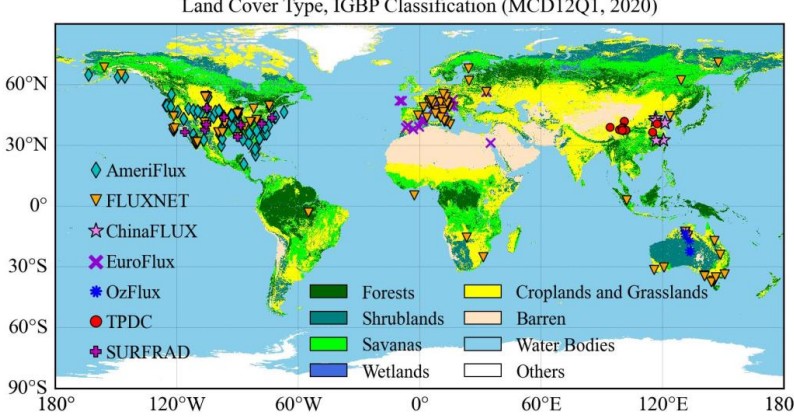


**Fig. 1 Spatial distribution of the 258 eddy covariance sites from AmeriFlux, FLUXNET,**
**EuroFlux, OzFlux, ChinaFLUX and TPDC, and nine radiation sites from SURFRAD involved**
**for analysis in this study.**
Furthermore, ground-based radiation observations from nine sites that are located
in large flat agricultural areas covered by crops and grasses from SURFRAD were also



introduced to validate land surface radiation estimates. Similar to the preprocessing
performed on the observations of the 258 EC sites, the $SW_{IN}$, $SW_{OUT}$, $LW_{IN}$, $LW_{OUT}$ and
Rn from the SURFRAD were also quality-controlled and aggregated to daily data.
Spatial distribution of the 258 EC sites and nine radiation sites from SURFRAD are
shown in Fig. 1, with site details (latitude, longitude, land cover types, digital elevation
model and temporal coverage) provided in Supplementary Tables S1 and S2.
**Table 1 Summary of mainstream datasets/products for inter-comparison used in this study**

| Products/ datasets | Reso-lution | Time coverage | Variables | Algorithms | References |
|---|---|---|---|---|---|
| GLASS | 0.05°/ daily | 2000-2018 | $SW_{IN}$, $LW_{IN}$, $LW_{OUT}$, Rn | Machine learning, direct estimation algorithm | Wang et al. (2015); Xu et al. (2022b); Jiang et al. (2015) |
| BESS-Rad | 0.05°/ daily | 2000-2020 | $SW_{IN}$ | BESS process model | Ryu et al. (2018) |
| BESSV2.0 | 0.05°/ daily | 2000-2020 | Rn, LE | BESS process model | Li et al. (2023) |
| FLUXCOM | 0.0833°/ 8-day | 2000-2020 | Rn, LE, H | Model tree ensembles | Jung et al. (2019) |
| MOD16A2 | 500 m/ 8-day | 2000-2020 | Rn, LE | Modified Penman-Monteith equation | Mu et al. (2011) |
| PML_V2 | 500 m/ 8-day | 2002-2020 | LE | Penman Monteith-Leuning model, Priestly Taylor equation and Gash model | Zhang et al. (2019) |
| ETMonitor | 1 km/ daily | 2000-2020 | LE | Shuttleworth-Wallace two-source scheme, Gash model and Penman equation | Zheng et al. (2022) |

**2.2 Climate/meteorology and remote sensing data**
To generate global datasets of land surface radiation and heat fluxes from 2000 to
2020, five types of climate/meteorology and remote sensing data were used in this study,
including:



(1)   ERA5-Land reanalysis datasets (https://cds.climate.copernicus.eu/, last access: 6

August 2024) with the spatial resolution of ~9 km from 1950 (Muñoz-Sabater et

al., 2021). Following our previous work (Wang et al., 2025), this study used

variables from the ERA5-Land datasets to drive the model, including near-surface

2 m air temperature ($T_a$), soil temperature in layer 1 (0-7 cm, $T_{S1}$), soil volumetric

moisture content in layer 1 (0-7 cm, $SM1$), solar radiation reaching the surface

of the earth ($SW_{IN}^{ERA5}$), net thermal radiation at the surface ($LW_{net}$), pressure of the

atmosphere ($PA$), 10 m wind speed ($WS$), precipitation ($P_r$) and the 2 m

dewpoint temperature, daily minimum and maximum air temperature [for

calculating relative air humidity ($RH$)].

(2)   GLASS datasets (https://glass.bnu.edu.cn/, last access: 6 August 2024), which

provide the 500 m 8-day leaf area index ($LAI$) and fractional vegetation cover

($FVC$) from February 2000 to December 2021.

(3)   MOD44B product (https://lpdaac.usgs.gov/, last access: 6 August 2024), which

offers yearly 250 m percent tree cover ($PTC$) since 2000, representing the

percentage (0~100%) of a pixel covered by tree canopy.

(4)   NOAA/GML atmospheric carbon dioxide ($CO_2$) concentration data, providing

monthly    global    marine    surface    mean    data    since    1958

(ftp://aftp.cmdl.noaa.gov/products/trends/co2/co2_mm_gl.txt,    last    access:    6

August 2024).

(5)   GMTED2010                    topographic                    data

(https://topotools.cr.usgs.gov/gmted_viewer/gmted2010_global_grids.php,    last

access: 6 August 2024), providing 500 m digital elevation model (DEM), slope,

and aspect.

The ~9 km ERA5-Land datasets were spatially interpolated to 500 m using the

cubic convolution method, and the 250 m PTC was resampled to 500 m using the
arithmetic averaging method.





### 2.3 Mainstream datasets/products for inter-comparison


Mainstream RS-based datasets/products of moderate-resolution global land

surface radiation and heat fluxes were collected for inter-comparison (Table 1),
including (1) the daily 0.05° GLASS $SW_{IN}$, $LW_{IN}$, $LW_{OUT}$ and Rn products from 2000 to
2018 (https://glass.bnu.edu.cn/, last access: 6 August 2024), (2) the daily 0.05°
Breathing Earth System Simulator Radiation (BESS-Rad) $SW_{IN}$ products from 2000 to
2020 (https://www.environment.snu.ac.kr/bess-rad), (3) the daily 0.05° BESS
Version2.0 (BESSV2.0) Rn and LE products from 2000 to 2020
(https://www.environment.snu.ac.kr/bessv2), (4) the 8-day 0.0833° FLUXCOM Rn, LE
and H products from 2001 to 2020 (https://fluxcom.org/, last access: 6 August 2024),
(5) the daily 1 km ETMonitor LE product from 2000 to 2020 (https://data.casearth.cn/,
last access: 6 August 2024), (6) the 8-day 500 m Penman-Monteith-Leuning Version2
(PML_V2, https://www.tpdc.ac.cn/, last access: 6 August 2024) LE product from 2000
to 2020; and (7) the 8-day 500 m MOD16A2 (https://lpdaac.usgs.gov/, last access: 6
August 2024) LE product from 2000 to 2020.

The GLASS $SW_{IN}$ products are derived from a combination of the GLASS

broadband albedo product and the surface shortwave net radiation estimates, where the
surface shortwave net radiation is estimated using linear regression with MODIS top-
of-atmosphere (TOA) spectral reflectance (Wang et al., 2015). The GLASS $LW_{IN}$ and
$LW_{OUT}$ products are generated using densely connected convolutional neural networks,
incorporating Advanced Very High-Resolution Radiometer (AVHRR) TOA reflectance
and ERA5 near-surface meteorological data (Xu et al., 2022b). The GLASS Rn
products are estimated from the meteorological variables from MERRA2 and surface
variables from GLASS using the multivariate adaptive regression splines model (Jiang
et al., 2015). The BESS-Rad and BESSV2.0 estimate $SW_{IN}$ and Rn using a radiative
transfer model (i.e. Forest Light Environmental Simulator, FLiES) with an artificial
neural network based on MODIS and MERRA2 reanalysis datasets, and using FLiES
based on MODIS products and NCEP/NCAR reanalysis data, respectively (Li et al.,



2023; Ryu et al., 2018). Moreover, the BESSV2.0 (Li et al., 2023), MOD16A2 (Mu et
al., 2011), PML_V2 (Zhang et al., 2019) and ETMonitor (Zheng et al., 2022) generated
global LE by physical models, such as Penman-Monteith equation, Priestley-Taylor
equation and/or Shuttleworth-Wallace two-source scheme. The FLUXCOM Rn, LE and
H datasets are obtained through multiple machine learning methods based on in situ
observations from FLUXNET and remote sensing and meteorological data (Jung et al.,
2019). For better consistency, RF-based 8-day 0.0833° Rn and Bowen ratio-corrected
LE and H for the periods of 2000 to 2020 from the FLUXCOM were used in this study.
**3 Methods**

The method used to generate global datasets of land surface radiation and heat

fluxes is based on the CoSEB model/framework, which was developed by our recently
published work (Wang et al., 2025) to coordinately estimate global land surface energy
balance components (including Rn, LE, H and G) using the multivariate random forest
technique, with a combination of MODIS and GLASS products, ERA5-Land reanalysis
datasets, and in situ observations at 336 EC sites from the FLUXNET, AmeriFlux,
ChinaFLUX, EuroFlux, OzFlux and Heihe River Basin flux network. The CoSEB
model was demonstrated to be able to produce high-accuracy estimates of land surface
energy components, with the RMSE of <17 W/m$^2$ for estimating 4-day Rn, LE and H,
and the RMSE of <5 W/m$^2$ for estimating 4-day G. The most praiseworthy superiority
of the CoSEB model lies in its ability to balance the land surface energy components,
with an energy imbalance ratio [EIR, defined as $100\% \times (Rn - G - LE - H)/Rn$] of 0.

To coordinately estimate land surface radiation and heat fluxes that comply with

both radiation balance and heat balance, one of the key procedures in the construction
of the CoSEB model was to prepare training datasets that satisfy surface radiation and
heat balance. For this purpose, the energy-imbalance corrections on daily in situ
observed LE and H were conducted by the most widely applied Bowen ratio method
[ $H^{corr} = \dfrac{H}{H + LE} \times (Rn - G)$, $LE^{corr} = \dfrac{LE}{H + LE} \times (Rn - G)$, where $H^{corr}$ and $LE^{corr}$



represent the sensible heat flux and latent heat flux after energy-imbalance correction,
respectively] with the aid of Rn and G observations, and the in situ Rn was calculated
from the sum of in situ observed net longwave radiation ($LW_{IN}$ minus $LW_{OUT}$) and net
shortwave radiation ($SW_{IN}$ minus $SW_{OUT}$). The input variables to renew the CoSEB
model include: (1) climate/meteorology: $T_a$, $SW_{IN}^{ERA5}$, $LW_{net}$, $WS$, $PA$, $P_r$, $RH$,
$CO_2$ concentration; (2) vegetation and soil: $LAI, FVC, PTC$, $T_{S1}$, $SM1$; (3) topography
data: $DEM$, $Slope$ and $Aspect$, in addition to longitude ($Lon$), latitude ($Lat$), and inverse
relative distance from the Earth to the Sun ($dr$), in which the $dr$ was calculated as
$dr = 1 + 0.033 \times \cos\left(\dfrac{2\pi \times DOY}{365}\right)$, where $DOY$ represents the day of year. Considering
that the footprint of the site-based measurements of turbulent heat fluxes is generally at
a scale of hundreds of meters, to reduce the effect of differences of spatial scales
between ground-based measurements (dependent variables) and remotely
sensed/reanalysis datasets (independent variables), we renewed the CoSEB model at a
spatial scale of 500 m for coordinately estimating global daily land surface radiation
and heat fluxes, which can be expressed as follows:
$$\begin{pmatrix} SW_{IN}, SW_{OUT}, LW_{IN}, \\ LW_{OUT}, Rn, LE, H, G \end{pmatrix} = f\begin{pmatrix} Lon, Lat, T_a, T_{S1}, SM1, SW_{IN}^{ERA5}, LW_{net}, PA, WS, P_r, dr \\ RH, LAI, FVC, PTC, DEM, Slope, Aspect, CO_2 \end{pmatrix} (1)$$
For comparison, eight RF-based uncoordinated models for separate estimates of
$SW_{IN}$, $SW_{OUT}$, $LW_{IN}$, $LW_{OUT}$, Rn, LE, H and G were also constructed using the same
inputs as those in the renewed CoSEB model. Site-based 10-fold cross-validation was
employed to assess the transferability and generalization of the CoSEB model by
randomly dividing all sites into ten folds, where each fold in turn serves as validation
datasets while the other folds as the training datasets, ensuring the validation of the
estimates of the CoSEB was conducted at sites that are spatially independent from those
selected for the training datasets. Fig. 2 illustrates the flowchart for generating global
datasets of land surface radiation and heat fluxes by the CoSEB model.

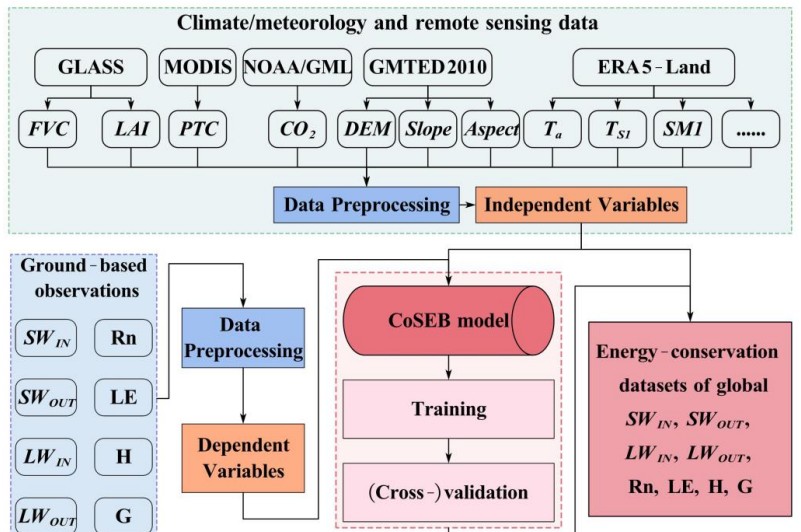


**Fig. 2 Flowchart for generating energy-conservation datasets of global land surface radiation**
**[including downward shortwave radiation ($SW_{IN}$), downward longwave radiation ($LW_{IN}$),**
**upward shortwave radiation ($SW_{OUT}$), upward longwave radiation ($LW_{OUT}$) and net radiation**
**(Rn)] and heat fluxes [including latent heat flux (LE), soil heat flux (G) and sensible heat flux**
**(H)] by the CoSEB model renewed from in situ observations at 258 sites worldwide and**
**collocated remote sensing and reanalysis datasets.**
**4 Results**
**4.1 Validation of the CoSEB model**
**4.1.1 Site-based 10-fold cross-validations at 258 EC sites**

Fig. 3 and Fig. 4 present the scatter density plots of the site-based 10-fold cross-

validation of daily $SW_{IN}$, $LW_{IN}$, $SW_{OUT}$, $LW_{OUT}$, Rn, LE, H and G estimated from the
renewed CoSEB model and the RF-based uncoordinated models, respectively, by using
the validation datasets collected at 258 EC sites worldwide. Results indicated that the
estimates from both the CoSEB model and the RF-based uncoordinated models agreed
well with the in situ observations, with the coefficient of determination ($R^2$) varying
between 0.80 and 0.95 for $SW_{IN}$, $LW_{IN}$, $LW_{OUT}$ and Rn, and between 0.59 and 0.67 for
$SW_{OUT}$, LE and H. The CoSEB model, with the root mean square error (RMSE) of 26.82
to 34.25 W/m² and mean absolute error (MAE) of 18.83 to 24.49 W/m² for $SW_{IN}$, Rn,
LE and H, the RMSE of 12.24 to 17.75 W/m² and the MAE of 8.39 to 13.70 W/m² for





$SW_{OUT}$, $LW_{IN}$ and $LW_{OUT}$, demonstrated comparable accuracies to the RF-based models,
with the RMSE of 27.07 to 33.34 W/m² and MAE of 19.29 to 23.64 W/m² for $SW_{IN}$,
Rn, LE and H, the RMSE of 12.12 to 16.93 W/m² and the MAE of 8.68 to 12.99 W/m²
for $SW_{OUT}$, $LW_{IN}$ and $LW_{OUT}$. In the validation of daily G, both the CoSEB and RF-based
models yielded RMSEs below 7 W/m². Strikingly, the CoSEB model exhibited large
superiority in balancing the surface radiation and heat fluxes, with the radiation
imbalance ratio [RIR, defined as $100\% \times \left( SW_{IN} + LW_{IN} - SW_{OUT} - LW_{OUT} - Rn \right) / Rn$ ]
and energy imbalance ratio [EIR, defined as $100\% \times \left( Rn - G - LE - H \right) / Rn$ ] of 0,
while the RF-based uncoordinated models showed substantial imbalances of the surface
radiation and heat fluxes, with RIR and EIR that were approximately normally
distributed, having absolute mean values of 38.84% and 31.22%, respectively, and
reaching as high as 50% in some cases.

It should be pointed out that the performances of both the renewed CoSEB model

and the RF-based models could be further improved if the site-based 10-fold cross-
validation was replaced with the sample-based 10-fold cross-validation (Figs. S1 and
S2 in the Supplementary Material). Specifically, for the CoSEB model, using the
sample-based 10-fold cross-validation decreased the RMSE by 0.61 to 3.92 W/m² for
five radiation components and G, and by 6.25 W/m² and 5.50 W/m² for LE and H,
respectively, in comparison to using the site-based 10-fold cross-validation. Likewise,
for the RF-based models, the RMSE decreased by 1.41 to 5.25 W/m² for five radiation
components and G, and by 9.63 W/m² and 7.43 W/m² for LE and H, respectively. The
$R^2$ of both the CoSEB model and the RF-based models using the sample-based 10-fold
cross-validation increased by 0.02 to 0.28 compared to the $R^2$ using the site-based 10-
fold cross-validation.

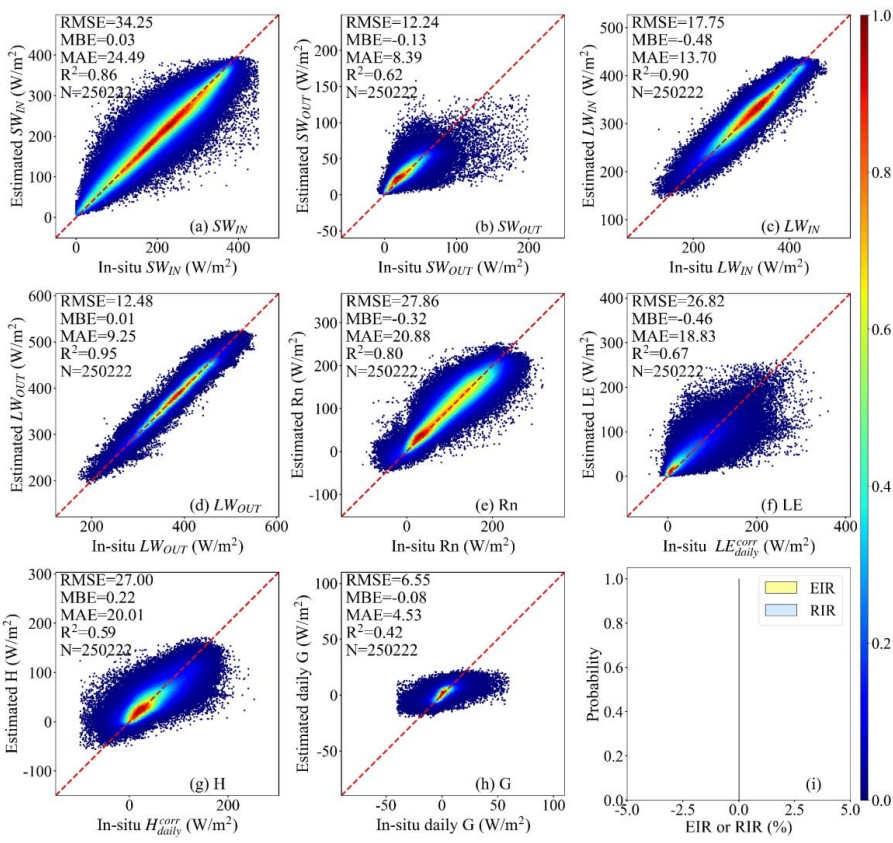

**Fig. 3 Scatter density plots of the site-based 10-fold cross-validation of daily downward shortwave and longwave radiation ($SW_{IN}$ and $LW_{IN}$), upward shortwave and longwave radiation ($SW_{OUT}$ and $LW_{OUT}$), net radiation (Rn), soil heat flux (G), latent heat flux (LE) and sensible heat flux (H) derived by the CoSEB model against in situ observed $SW_{IN}$, $LW_{IN}$, $SW_{OUT}$, $LW_{OUT}$, Rn, G, and energy imbalance-corrected LE ( $LE_{daily}^{corr}$ ) and H ( $H_{daily}^{corr}$ ). The EIR and RIR in the subfigure (i) represent the energy imbalance ratio and radiation imbalance ratio, which are defined as $100\% \times \left( Rn - G - LE - H \right) / Rn$ and $100\% \times \left( SW_{IN} + LW_{IN} - SW_{OUT} - LW_{OUT} - Rn \right) / Rn$ , respectively. The colorbar represents the normalized density of data points.**

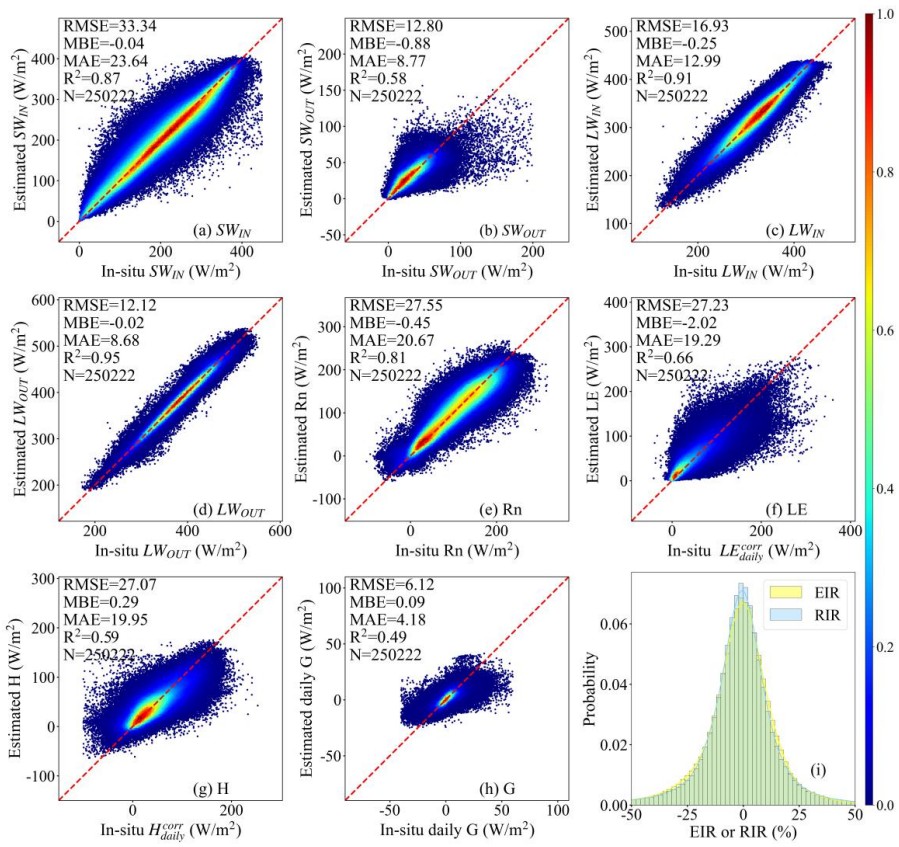

**Fig. 4 Same as Fig. 3, but for estimates from RF-based uncoordinated models.**
**4.1.2 Validation at nine radiation sites from SURFRAD**
To further illustrate the generality and transferability of the renewed CoSEB model,
the validation of estimates of the five radiation components (including $SW_{IN}$, $SW_{OUT}$,
$LW_{IN}$, $LW_{OUT}$, Rn) derived from both the CoSEB model and RF-based uncoordinated
models against observations at nine radiation sites from SURFRAD was performed, as
shown in Fig. 5. The results showed that both the CoSEB model and the RF-based
models achieved high accuracy in estimating daily $SW_{IN}$, $SW_{OUT}$, $LW_{IN}$, $LW_{OUT}$ and Rn,
with the RMSE of ~30 W/m$^2$ for $SW_{IN}$, ~14 W/m$^2$ for $SW_{OUT}$ and $LW_{IN}$, ~12 W/m$^2$ for
$LW_{OUT}$ and ~24 W/m$^2$ for Rn, with the R$^2$ >0.9 for $SW_{IN}$, $LW_{IN}$ and $LW_{OUT}$, ~0.65 for
$SW_{OUT}$ and ~0.85 for Rn. Compared to the results of the site-based 10-fold cross-
validation at 258 EC sites, the performances at nine radiation sites showed slight



improvements, with the RMSE decreasing by 0.74 to 4.54 W/m² for $SW_{IN}$, $LW_{IN}$, $LW_{OUT}$
and Rn in the CoSEB model, but a slight degradation with the RMSE increasing by
~1.05 W/m² for $SW_{OUT}$, suggesting the robust performance of the CoSEB model.
Furthermore, the CoSEB model demonstrated a large superiority in maintaining surface
radiation balance among the five radiation components, with the RIR of 0, in contrast
to the RF-based models, which failed to meet this balance, exhibiting significant RIR
exceeding 50%.

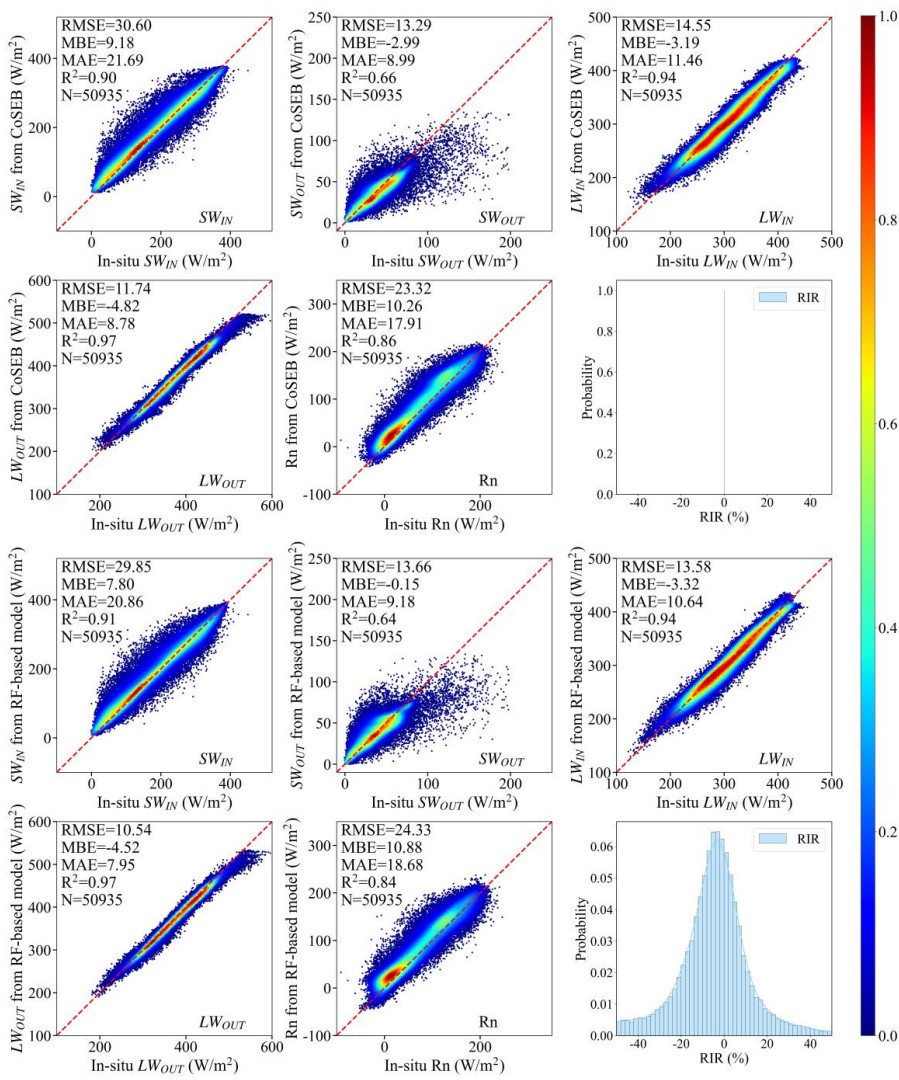

**Fig. 5 Scatter density plots of the validation of daily downward shortwave and longwave radiation ($SW_{IN}$ and $LW_{IN}$), upward shortwave and longwave radiation ($SW_{OUT}$ and $LW_{OUT}$) and net radiation (Rn) from the CoSEB-based datasets against in situ observations at nine radiation sites from SURFRAD. The RIR represents the radiation imbalance ratio, defined as**

$$100\% \times \left( SW_{IN} + LW_{IN} - SW_{OUT} - LW_{OUT} - Rn \right) / Rn$$ **. The colorbar represents the normalized density of data points.**

**4.2 Validation and inter-comparisons of the CoSEB-based datasets**

As demonstrated in Section 4.1, the renewed CoSEB model with a spatial scale of 500 m achieved comparable accuracies to the RF-based uncoordinated models but

outperformed them in balancing surface radiation and heat fluxes. Evidenced by the
validation for its superiority, the renewed CoSEB model was then applied to the
spatially aggregated input datasets to generate our developed global daily datasets with
a spatial resolution of 0.05°. To further assess the performance of the developed datasets,
in situ observations at 134 sites out of the 258 EC sites were further used to test the
performance of the CoSEB-based datasets, where the 134 sites were selected based on
the commonly applied criterion (Salazar-Martínez et al., 2022; Tang et al., 2024a) that
the fraction of the dominant land cover types (from the 500 m MCD12Q1 product)
exceeded 80% within the 0.05° grid, ensuring surface homogeneity and spatial
representativeness of the observations. Mainstream products (i.e. GLASS, BESS-Rad,
BESSV2.0, FLUXCOM, PML_V2, MOD16A2 and ETMonitor) were also involved for
inter-comparison at the 134 EC sites.
Note that due to the lack of moderate-resolution global RS-based products/datasets
of daily and/or 8-day $SW_{OUT}$, H and G, the intercomparison between different
products/datasets was impossible. Instead, we conducted a validation of these
components from the CoSEB-based datasets against in situ observations at 134 EC sites,
as shown in Figs S3 and S4 in the Supplementary Material. Results indicated that the
CoSEB-based datasets could provide good estimates of $SW_{OUT}$, H and G, with the
RMSE of 10.39 W/m², 22.67 W/m² and 6.77 W/m² at daily scale, respectively, and the
RMSE of 7.08 W/m² and 4.25 W/m² for 8-day $SW_{OUT}$ and G, respectively.
Fig. 6 and Fig. 7 present the comparison of daily $SW_{IN}$, $LW_{IN}$ and $LW_{OUT}$, as well
as Rn and LE from the CoSEB-based datasets and mainstream products/datasets
(including GLASS, BESS-Rad, BESSV2.0 and ETMonitor), with in situ observations
at 134 EC sites, respectively. Overall, the estimates from the CoSEB-based datasets
exhibited a closer agreement with in situ observations than those from mainstream
products/datasets, where the CoSEB-based datasets reduced the RMSE by 4.35 W/m²
to 11.46 W/m² and increased the R² by 0.04 to 0.3 compared to mainstream products.
Specifically, the RMSE for the $SW_{IN}$, $LW_{IN}$, $LW_{OUT}$ increased from 28.51 W/m², 14.29



W/m$^2$ and 10.62 W/m$^2$ in the CoSEB-based datasets to 35.44 W/m$^2$ ,18.64 W/m$^2$ and
15.29 W/m$^2$ in the GLASS, respectively, and for $SW_{IN}$ from 28.51 W/m$^2$ in the CoSEB-
based datasets to 36.23 W/m$^2$ in the BESS-Rad. Likewise, the RMSEs for daily Rn and
LE were 22.40 W/m$^2$ and 24.38 W/m$^2$ in the CoSEB-based datasets, which were lower
than those of 29.80 W/m$^2$ and 35.75 W/m$^2$ in BESSV2.0, respectively, as well as those
of 27.11 W/m$^2$ for Rn in GLASS and 35.84 W/m$^2$ for LE in ETMonitor.

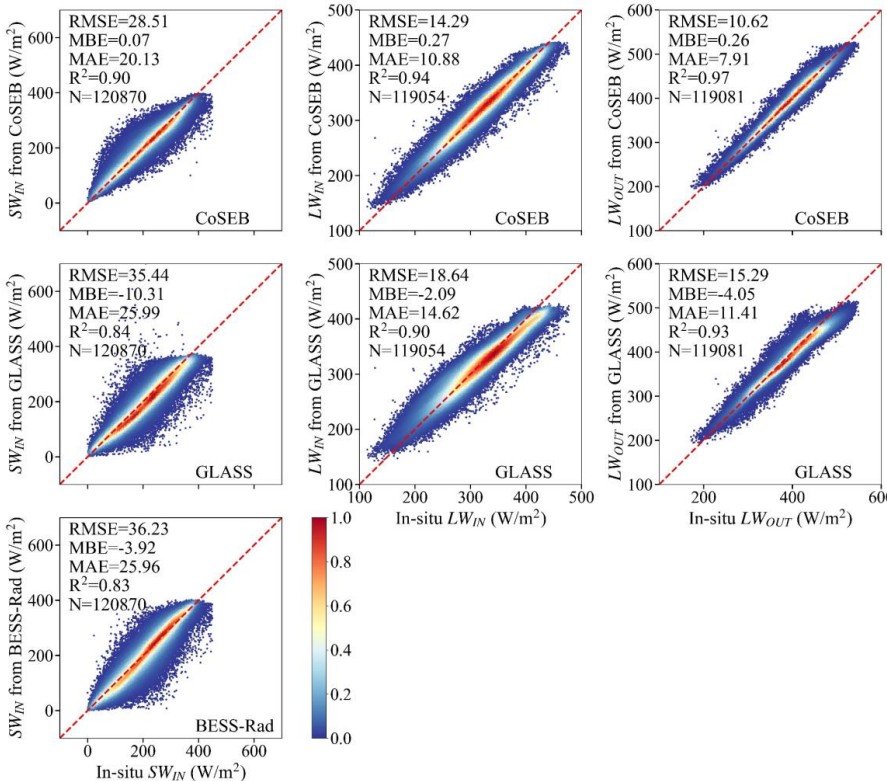


**Fig. 6 Comparison of the daily downward shortwave radiation ($SW_{IN}$, the first column),**
**downward longwave radiation ($LW_{IN}$, the second column) and upward longwave radiation**
**($LW_{OUT}$, the third column) from the CoSEB-based datasets, GLASS and BESS-Rad with the**
**in situ observed $SW_{IN}$, $LW_{IN}$ and $LW_{OUT}$ at 134 eddy covariance sites. The colorbar represents**
**the normalized density of data points.**

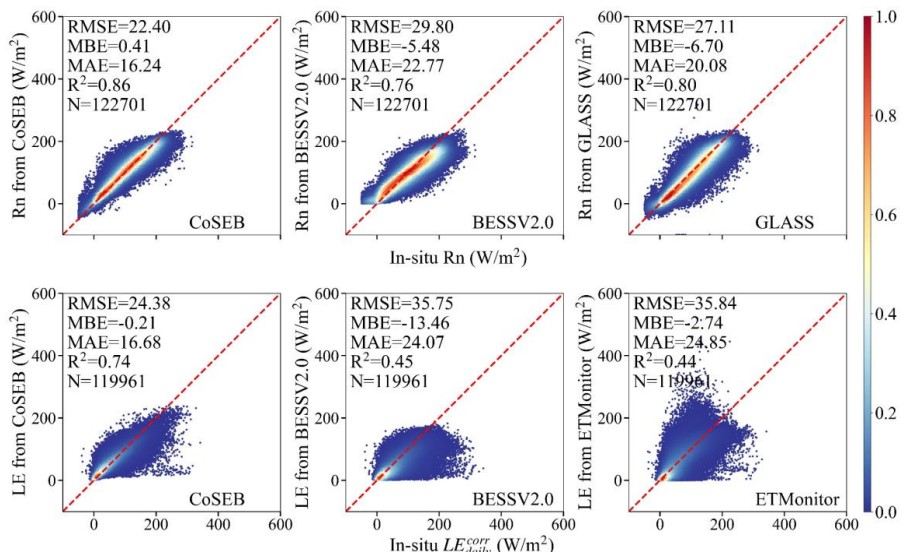

**Fig. 7 Comparison of the daily net radiation (Rn, the upper row) and latent heat flux (LE, the lower row) from the CoSEB-based datasets, BESSV2.0, GLASS and ETMonitor with the in situ observed Rn, and energy imbalance-corrected LE ( $LE_{daily}^{corr}$ ) at 134 eddy covariance sites. The colorbar represents the normalized density of data points.**

Figs. 8, 9 and 10 compare the 8-day $SW_{IN}$, $LW_{IN}$ and $LW_{OUT}$, Rn and LE, as well as H from the CoSEB-based datasets and mainstream products, with in situ observations at 134 EC sites, respectively. Overall, the CoSEB-based datasets outperformed the mainstream products/datasets for all surface radiation and heat fluxes, where the CoSEB-based datasets reduced the RMSE by 4.62 W/m² to 14.64 W/m² and increased the R² by 0.04 to 0.41 compared to mainstream products. Specifically, for $SW_{IN}$, $LW_{IN}$ and $LW_{OUT}$, the RMSE increased from 12.81 W/m², 9.22 W/m² and 8.34 W/m² in the CoSEB-based datasets to 21.23 W/m², 15.37 W/m² and 14.70 W/m² in the GLASS, respectively, and for $SW_{IN}$ from 12.81 W/m² in the CoSEB-based datasets to 17.43 W/m² in the BESS-Rad. For Rn, the RMSE increased from 13.38 W/m² in the CoSEB-based datasets to 18.64 W/m² in the GLASS and to >23 W/m² in the FLUXCOM and BESSV2.0, while the R² decreased from 0.91 in the CoSEB to 0.82 in the GLASS and to <0.72 in the FLUXCOM and BESSV2.0. Likewise, for LE, the RMSE increased from 19.99 W/m² in the CoSEB-based datasets to 26.16 W/m² in the FLUXCOM, and





to >28.17 W/m$^2$ in BESSV2.0, MOD16A2, PML_V2 and ETMonitor, while the R$^2$
decreased from 0.8 in the CoSEB-based datasets to 0.65 in the FLUXCOM, and to <0.6
in the remaining products. For H, the RMSE increased from 17.44 W/m$^2$ in the CoSEB-
based datasets to 23.96 W/m$^2$ in the FLUXCOM.

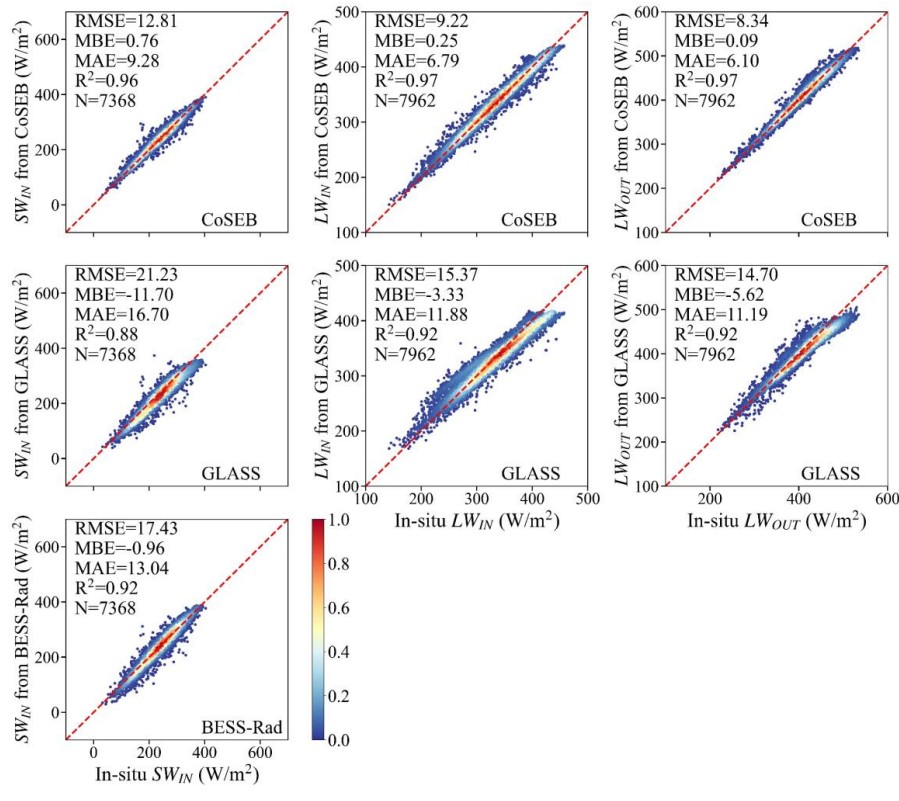


**Fig. 8 Same as Fig. 6, but for the comparison at 8-day scale.**



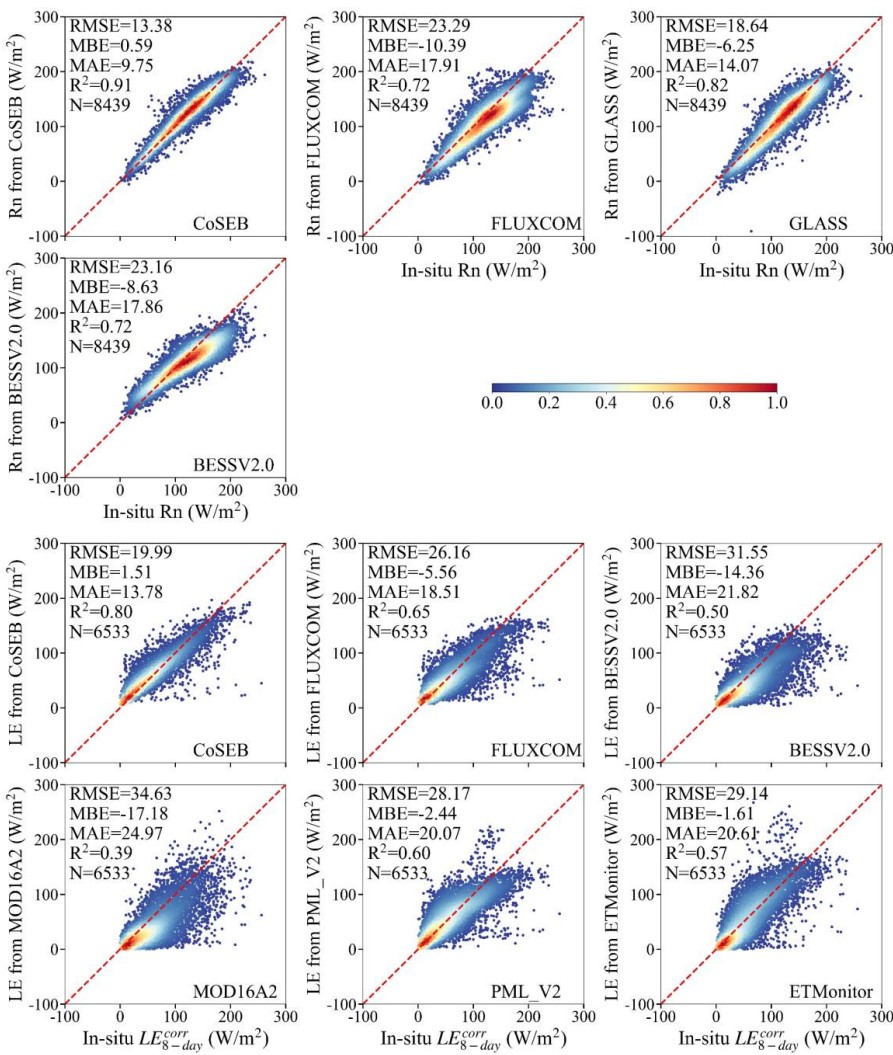

**Fig. 9 Comparison of the 8-day net radiation (Rn, the upper two rows) and latent heat flux (LE, the lower three rows) from the CoSEB-based datasets, FLUXCOM, BESSV2.0, GLASS, MOD16A2, PML_V2 and ETMonitor with in situ observed Rn, and energy imbalance-corrected LE ($LE_{8-day}^{corr}$) at 134 eddy covariance sites. The colorbar represents the normalized density of data points.**


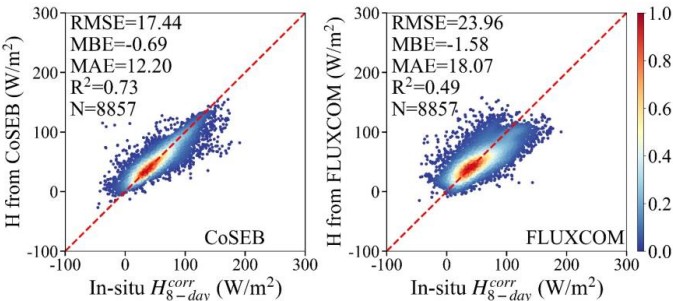

**Fig. 10 Comparison of the 8-day sensible heat flux (H) from the CoSEB-based datasets and**

**the FLUXCOM with the in situ energy imbalance-corrected H ( $H_{8-day}^{corr}$ ) at 134 eddy covariance**

**sites. The colorbar represents the normalized density of data points.**

**4.3 Spatial-temporal patterns of global land surface radiation and heat fluxes**

In addition to the validation and inter-comparison of the CoSEB-based datasets at

global sites, we further inter-compared the estimates of land surface radiation and heat

fluxes from the CoSEB-based datasets and the mainstream products/datasets, in terms

of their spatial and temporal patterns.

Figs. 11, 12 and 13 show the spatial distributions (excluding Greenland, Antarctic

continent, deserts, water bodies and permanent snow) and latitudinal profiles of the

global 0.05° mean annual $SW_{IN}$, $LW_{IN}$ and $LW_{OUT}$, Rn and LE, as well as H from 2001

to 2018, respectively, as derived from the CoSEB-based datasets and mainstream

products/datasets [i.e. GLASS, BESS-Rad, BESSV2.0, FLUXCOM, MOD16A2,

PML_V2 and ETMonitor, resampled to 0.05° using arithmetic averaging method or

cubic convolutional method if necessary]. Overall, the spatial patterns of the estimates

from the CoSEB-based datasets aligned well with those observed in these mainstream

products/datasets, though regional discrepancies were present. Specifically, the mean

annual $LW_{IN}$, $LW_{OUT}$, Rn, and LE generally exhibited decreasing trends from the equator

towards higher latitudes, peaking in regions such as the Amazon Rainforest, Congo

Rainforest, and the Malay Archipelago. In contrast, the higher mean annual $SW_{IN}$ and

H were mainly found in the Tibetan Plateau, southwestern U.S., mid-west Australia,

Sahel and Southern Africa, while the lower values were found in high-latitude regions





of >50°N. In the region of high values, the mean annual estimates of $SW_{IN}$ from the
CoSEB-based datasets were higher than those from GLASS but lower than those from
BESS-Rad, the estimates of $LW_{IN}$ and $LW_{OUT}$ from the CoSEB-based datasets were both
higher than those from GLASS, the estimates of Rn from the CoSEB-based datasets
were significantly higher than those from BESSV2.0, and comparable to or slightly
higher than those from FLUXCOM and GLASS, the estimates of LE from the CoSEB-
based datasets were close to those from BESSV2.0 and PML_V2, but slightly lower
than those from FLUXCOM, MOD16A2 and ETMonitor. Besides, the estimates of H
from the CoSEB-based datasets were higher than those from FLUXCOM in regions
with high values, while lower than those from FLUXCOM in regions with low values.



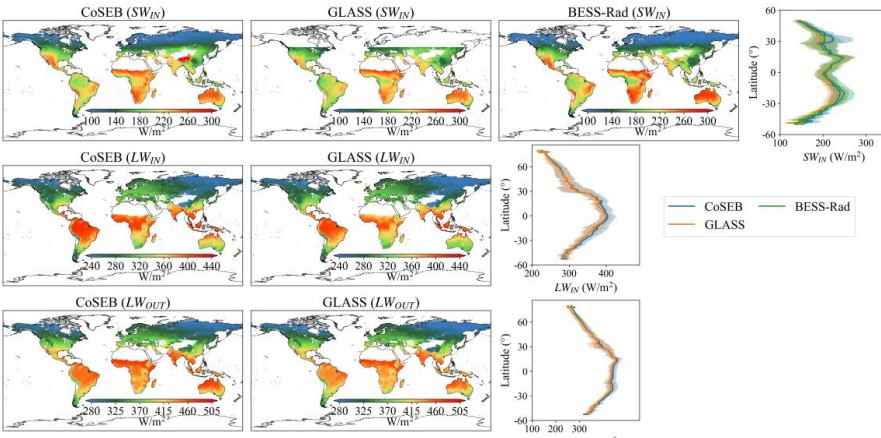


**Fig.11 Spatial patterns of global mean annual downward shortwave radiation ($SW_{IN}$, the first row), downward longwave radiation ($LW_{IN}$, the second row) and upward longwave radiation ($LW_{OUT}$, the third row) from 2001 to 2018 by CoSEB-based datasets, GLASS and BESS-Rad. The rightmost subfigure of each row represents the latitudinal profiles of mean annual $SW_{IN}$, $LW_{IN}$ and $LW_{OUT}$ from CoSEB-based datasets, GLASS and BESS-Rad, where the shaded area represents the variation of standard deviation for each product.**

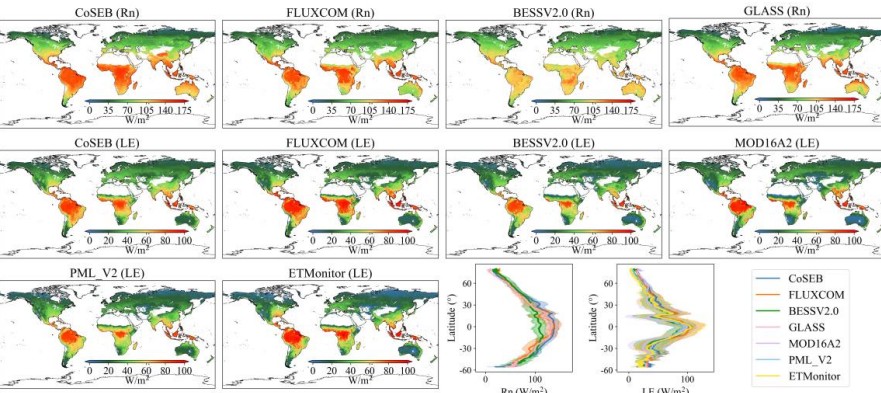

**Fig.12 Spatial patterns of global mean annual net radiation ($Rn$, the first row) and latent heat flux ($LE$, the second and third rows) from 2001 to 2018 by CoSEB-based datasets, FLUXCOM, BESSV2.0, MOD16A2, PML_V2, ETMonitor and GLASS. The last two subfigures of the third row represent the latitudinal profiles of mean annual Rn and LE from CoSEB-based datasets and these mainstream products/datasets, where the shaded area represents the variation of standard deviation for each product.**

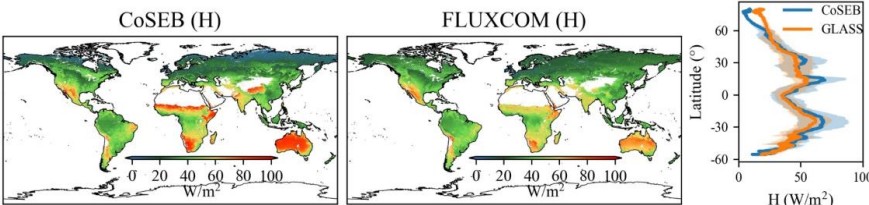

**Fig.13 Spatial patterns of global mean annual sensible heat flux (*H*) from 2001 to 2018 by CoSEB-based datasets and FLUXCOM. The rightmost subfigure represents the latitudinal profiles of mean annual H from CoSEB-based datasets and FLUXCOM, where the shaded area represents the variation of standard deviation for each product.**

The temporal evolutions of the global (excluding Greenland, Antarctic continent,
deserts, water bodies and permanent snow) land surface radiation and heat fluxes
derived from the CoSEB-based datasets and mainstream products/datasets from 2001
to 2018 were also investigated, as shown in Fig. 14. The results indicated that the
temporal variation of each flux from the CoSEB-based datasets generally agreed well
with those from mainstream products/datasets, exhibiting relatively stable trends. The
global annual mean estimates using area weighting average by the CoSEB-based
datasets from 2001 to 2018 varied between ~185.22 and ~189.50 W/m$^2$ with the mean
of ~187.23 W/m$^2$ for $SW_{IN}$, between ~32.67 and ~33.20 W/m$^2$ with the mean of ~32.96
W/m$^2$ for $SW_{OUT}$, between ~330.24 and ~334.14 W/m$^2$ with the mean of ~331.50 W/m$^2$
for $LW_{IN}$, between ~387.25 and ~390.82 W/m$^2$ with the mean of ~388.81 W/m$^2$ for
$LW_{OUT}$, between ~95.41 and ~99.39 W/m$^2$ with the mean of 97.11 W/m$^2$ for Rn,
between ~53.24 and ~56.37 W/m$^2$ with the mean of ~54.53 W/m$^2$ for LE, between
~40.44 and ~41.96 W/m$^2$ with the mean of ~41.29 W/m$^2$ for H, and between ~1.22 and
~1.52 W/m$^2$ with the mean of ~1.33 W/m$^2$ for G. For each radiation or heat flux, the
annual mean estimates from the CoSEB-based datasets were overall higher than those
from the mainstream products/datasets. In particular, the annual mean Rn estimates
from the CoSEB-based datasets were higher than those from FLUXCOM, GLASS and
BESSV2.0 sequentially, and the annual mean LE estimates from the CoSEB-based
datasets were marginally higher than those from FLUXCOM, but substantially
exceeded those from ETMonitor, PML_V2, MOD16A2 and BESSV2.0 sequentially.

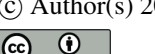

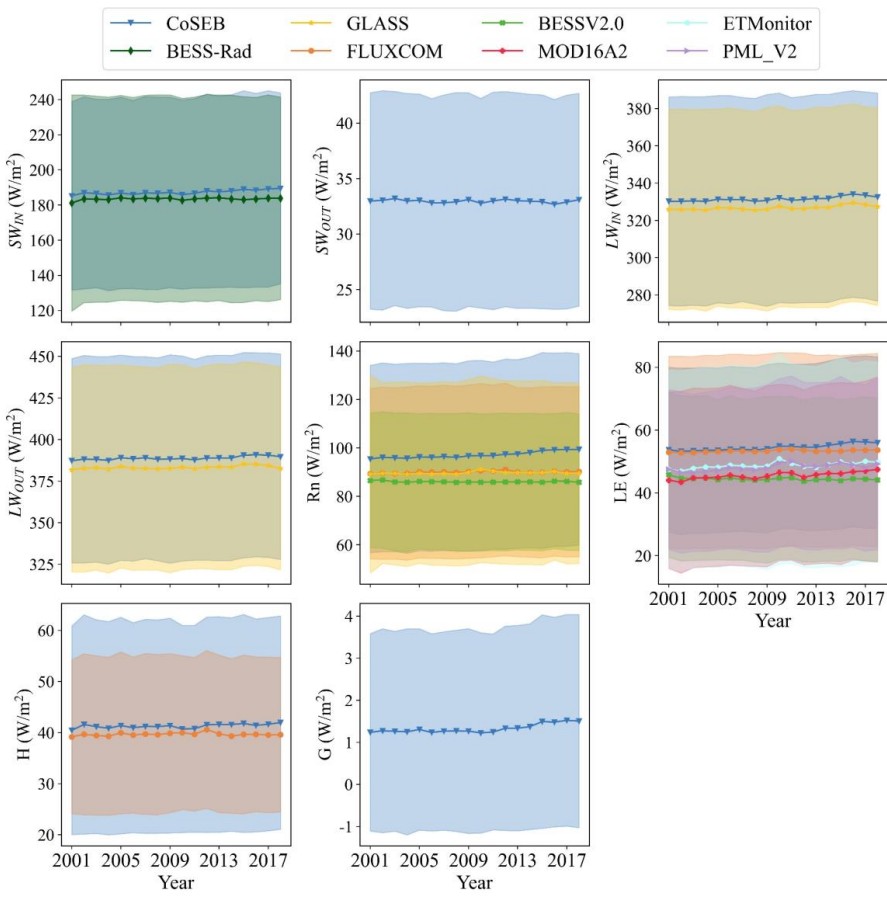


**Fig. 14 Temporal variation of annual mean downward shortwave radiation ($SW_{IN}$), upward shortwave radiation ($SW_{OUT}$), downward longwave radiation ($LW_{IN}$), upward longwave radiation ($LW_{OUT}$), net radiation (Rn), latent heat flux (LE), sensible heat flux (H) and soil heat flux (G) from 2001 to 2018 from the CoSEB-based datasets, BESS-Rad, GLASS, FLUXCOM, BESSV2.0, PML_V2, MOD16A2 and ETMonitor. The shaded area represents the variation of standard deviation for each product.**

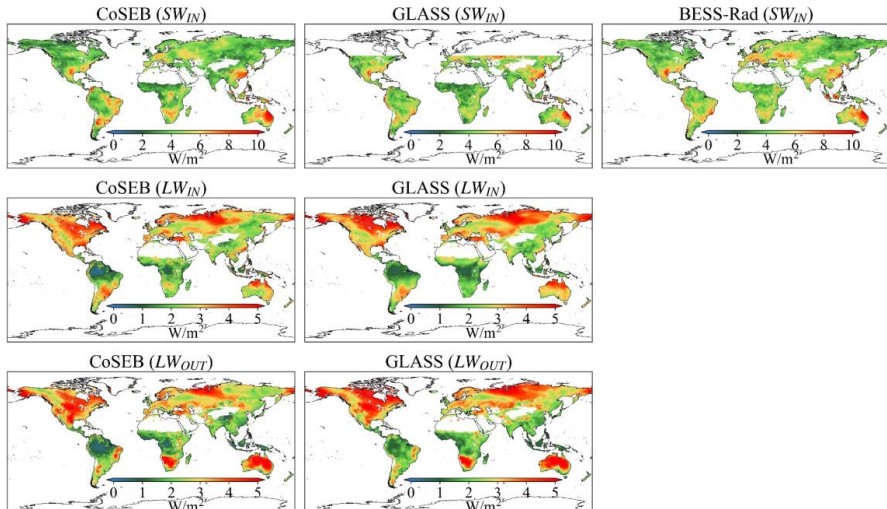

**Fig. 15 Spatial distribution of interannual variability (standard deviation) of downward shortwave radiation *(SW$_{IN}$*, the first row), downward longwave radiation (*LW$_{IN}$*, the second row) and upward longwave radiation (*LW$_{OUT}$*, the third row) from 2001 to 2018 by the CoSEB-based datasets, GLASS and BESS-Rad.**

520   Figs. 15, 16 and 17 show the spatial patterns (excluding Greenland, Antarctic

521  continent, deserts, water bodies and permanent snow) of interannual variability of *SW$_{IN}$*,

522  *LW$_{IN}$* and *LW$_{OUT}$*, Rn and LE, as well as H from 2001 to 2018, respectively, derived

523  from the CoSEB-based datasets and mainstream products/datasets. In general, the

524  estimates from the CoSEB-based datasets displayed similar interannual variability in

525  space with those from the mainstream products/datasets. Specially, the estimates of

526  *SW$_{IN}$* from the CoSEB-based datasets, BESS-Rad, and GLASS exhibited a significant

527  interannual variability mainly in northeastern Australia, eastern South America,

528  Southeast China, and Southwest North America. The interannual variability of *LW$_{IN}$*

529  and *LW$_{OUT}$* by the CoSEB-based datasets and GLASS displayed high values primarily

530  at middle-to-high latitudes of the North Hemisphere and parts of Africa and Australia.

531  The interannual variability of Rn observed by the CoSEB-based datasets was generally

532  lower than that of GLASS, but higher than that of BESSV2.0 and FLUXCOM. The

533  CoSEB-based datasets missed the strong interannual variability of LE as observed in

534  MOD16A2, PML_V2 and ETMonitor in parts of Africa, Australia and eastern South

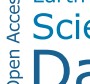

America. Furthermore, FLUXCOM exhibited the weakest interannual variability of LE
in almost all regions. The interannual variability of H derived from the CoSEB-based
datasets was higher than those from FLUXCOM, with stronger interannual variabilities
mainly observed in parts of eastern South America, southern Africa, and northeastern
Australia.

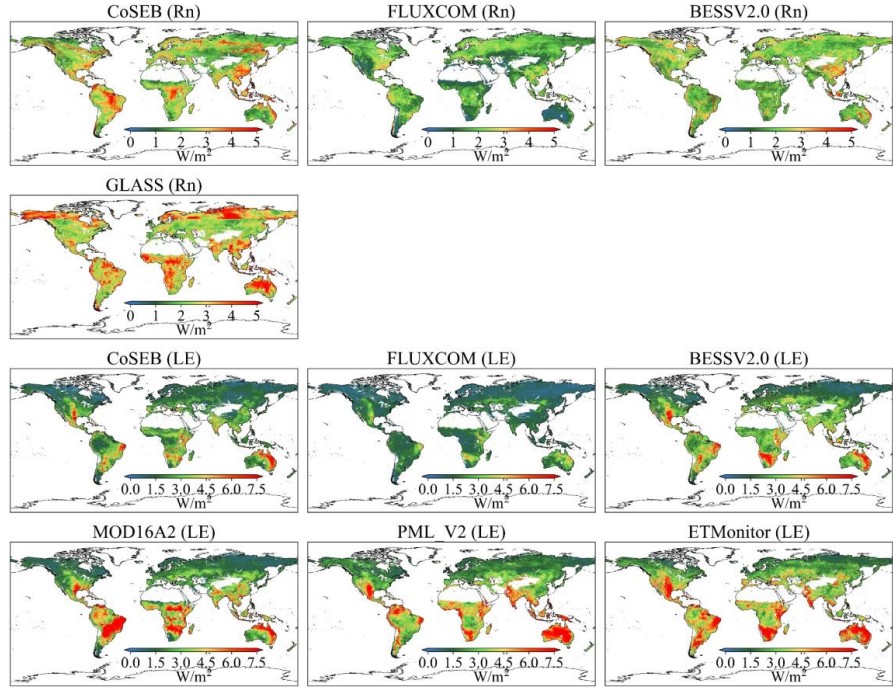

**Fig. 16 Spatial distribution of interannual variability (standard deviation) of net radiation (Rn,**
**the first and second rows) and latent heat flux (LE, the third and fourth row) from 2001 to**
**2018 by the CoSEB-based datasets, FLUXCOM, BESSV2.0, MOD16A2, PML_V2,**
**ETMonitor and GLASS.**

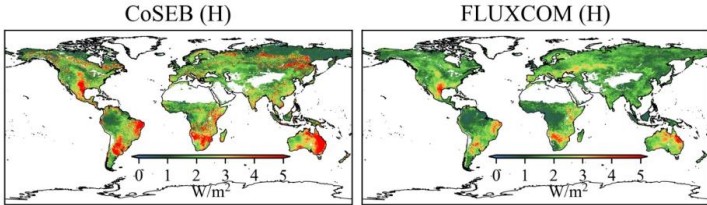

**Fig. 17 Spatial distribution of interannual variability (standard deviation) of sensible heat flux**
**(H) from 2001 to 2018 by the CoSEB-based datasets and FLUXCOM.**





**5 Discussion**


Accurately monitoring the spatial and temporal variations of global land surface
radiation and heat fluxes is crucial for quantifying the exchange of radiation, heat and
water between the land and atmosphere under global climate change (Chen et al., 2020;
Du et al., 2024; Kim et al., 2023; Liang et al., 2006; Wang et al., 2020). However,
although numerous global RS-based products/datasets of land surface radiation and
heat fluxes have been developed using physical and/or statistical methods, they
typically provide either merely a single flux or multiple fluxes (see Table 1) that are
estimated separately from uncoordinated models (Huang et al., 2024; Jung et al., 2019;
Sun et al., 2023; Tang et al., 2019), leading to noticeable radiation imbalance and/or
heat imbalance when these products are combined for practical applications. To address
these limitations, we generated high-accuracy global datasets of land surface radiation
and heat fluxes from 2000 to 2020 that adhere to both radiation and heat conservation
laws, using our proposed CoSEB model (Wang et al., 2025).
Our CoSEB model, integrating underlying physical principles of training datasets
into machine learning technique to effectively learn the interrelations among multiple
targeted outputs, was originally designed for coordinating estimates of global land
surface energy balance components (Rn, LE, H and G) to satisfy the energy
conservation (Wang et al., 2025). Inspired by the idea of constructing the CoSEB model,
we further incorporated land surface radiation fluxes into our model to simultaneously
consider the physical constraints of both surface radiation and heat conservation
principles, by renewing the CoSEB using remote sensing products, reanalysis datasets,
as well as in situ observations of $SW_{IN}$, $SW_{OUT}$, $LW_{IN}$, $LW_{OUT}$, Rn, LE, H and G. To
comprehensively account for the main factors influencing surface radiation and heat
fluxes (Amani & Shafizadeh-Moghadam, 2023; Mohan et al., 2020; Wang et al., 2021),
the renewed CoSEB model utilized 19 easily accessible parameters/variables from
ERA5-Land reanalysis datasets, GLASS products, MODIS products, GMTED2010 and
NOAA/GML as input, which were readily available to generate datasets of global land



surface radiation and heat fluxes in a practical and operational manner.
The main advantages of our CoSEB-based datasets of land surface radiation and
heat fluxes lie in that [1] they are the first RS-based global datasets that satisfy both
surface radiation balance ( $SW_{IN} - SW_{OUT} + LW_{IN} - LW_{OUT} = Rn$ ) and heat balance
( $LE + H + G = Rn$ ) among the eight fluxes, as demonstrated by both the RIR and EIR
of 0, [2] the radiation and heat fluxes are characterized by high accuracies when
validated against in situ measurements at 134 "homogeneous" sites (see the first
paragraph in Section 4.2), where (1) the RMSEs for daily estimates of $SW_{IN}$, $SW_{OUT}$,
$LW_{IN}$, $LW_{OUT}$, Rn, LE, H and G from the CoSEB-based datasets were 28.51 W/m², 10.39
W/m², 14.29 W/m², 10.62 W/m², 22.40 W/m², 24.38 W/m², 22.67 W/m² and 6.77 W/m²,
respectively, as well as for 8-day estimates were 12.81 W/m², 7.08 W/m², 9.22 W/m²,
8.34 W/m², 13.38 W/m², 19.99 W/m², 17.44 W/m² and 4.25 W/m², respectively, (2) the
CoSEB-based datasets, in comparison to the mainstream products/datasets (i.e. GLASS,
BESS-Rad, FLUXCOM, BESSV2.0, MOD16A2, PML_V2 and ETMonitor), better
agreed with the in situ observations at 134 EC sites, showing the RMSE reductions
ranging from 4.35 W/m² to 11.46 W/m² for $SW_{IN}$, $LW_{IN}$, $LW_{OUT}$, Rn and LE at daily
scale, and 4.62 W/m² to 14.64 W/m² for $SW_{IN}$, $LW_{IN}$, $LW_{OUT}$, Rn, LE and H at 8-day
scale.
Our developed datasets could be potentially applied in many fields, including but
not limited to (1) exploring the spatial-temporal patterns of global land surface radiation
and heat flux (es) and their driving mechanisms over the past decades under global
change (e.g., rising $CO_2$ concentration, greening land surface and increasing air
temperature), (2) investigating the variability of land surface radiation and heat fluxes
caused by extreme events and human activities, e.g. afforestation or deforestation,
wildfire, air pollution, weather extremes and urbanization, (3) assessing the resources
of solar energy, geothermal energy, surface and ground water at regional and global
scales, (4) monitoring natural hazards, e.g. drought in agriculture and forestry.
The uncertainties of our datasets are relevant to (1) the data preprocessing, and (2)





the application of the CoSEB at different spatial scales. Specifically, the daily average
of surface radiation and heat fluxes for each day was obtained for analysis from good-
quality half-hourly observations when the fraction of these good-quality half-hourly
observations was greater than 80% in a day, due to the lack of consensus on the method
for aggregating gapped half-hourly observations to daily data (Tang et al., 2024a; Yao
et al., 2017; Zheng et al., 2022). Likewise, since there was no agreement on how to
correct for the energy imbalance of turbulent heat fluxes, we adopted the most widely
applied Bowen ratio method to enforce energy closure between $Rn - G$ and $LE + H$
(Castelli et al., 2018; Twine et al., 2000; Zhang et al., 2021). These data preprocessing
had an effect on the construction of the renewed CoSEB model, which may further
affect the global datasets. Moreover, the renewed CoSEB model was constructed at the
spatial scale of 500 m to match the footprints of the in situ EC observations, but applied
at the spatial resolution of 0.05° to generate global datasets, mainly limited by the
computing and storage capabilities in our personal computers. However, the CoSEB-
based datasets have also been validated and inter-compared at 134 EC sites to
demonstrate that the difference in spatial scale would not much affect the performance
of the datasets. Despite these uncertainties, it is worth emphasizing that our work was
the first attempt to innovatively develop energy-conservation datasets of global land
surface radiation and heat fluxes with high accuracies.
**6 Data availability**

The energy-conservation datasets of global land surface radiation and heat fluxes

generated by the CoSEB model with spatial-temporal resolutions of daily and 0.05°
from Feb.26, 2000 to Dec.31, 2020 are freely available through the National Tibetan
Plateau Data Center at https://doi.org/10.11888/Terre.tpdc.302559 (Tang et al., 2025a)
and through the Science Data Bank (ScienceDB) at
https://doi.org/10.57760/sciencedb.27228 (Tang et al., 2025b).



**7 Summary and Conclusion**
This study for the first time developed energy-conservation datasets of global land
surface radiation and heat fluxes using our CoSEB model renewed based on GLASS
and MODIS products, ERA5-Land reanalysis datasets, topographic data, $CO_2$
concentration data, and observations at 258 EC sites worldwide from the FLUXNET,
AmeriFlux, EuroFlux, OzFlux, ChinaFLUX and TPDC.
The CoSEB-based datasets of land surface radiation and heat fluxes are the first
RS-based global datasets that satisfy both surface radiation balance
($SW_{IN} - SW_{OUT} + LW_{IN} - LW_{OUT} = Rn$) and heat balance ($LE + H + G = Rn$) among the
eight fluxes. Meanwhile, the CoSEB-based datasets outperformed the mainstream
products/datasets in accuracy. Specifically, at 134 EC sites, the RMSEs for daily
estimates of $SW_{IN}$, $SW_{IN}$, $LW_{IN}$, $LW_{OUT}$, Rn, LE, H and G from the CoSEB-based datasets
were 28.51 W/m², 10.39 W/m², 14.29 W/m², 10.62 W/m², 22.40 W/m², 24.38 W/m²,
22.67 W/m² and 6.77 W/m², respectively, as well as for 8-day estimates were 12.81
W/m², 7.08 W/m², 9.22 W/m², 8.34 W/m², 13.38 W/m², 19.99 W/m², 17.44 W/m² and
4.25 W/m², respectively. Moreover, the estimates from the CoSEB-based datasets in
comparison to those from the mainstream products/datasets reduced the RMSE by 4.35
W/m² to 11.46 W/m² and increased the R² by 0.04 to 0.3 for $SW_{IN}$, $LW_{IN}$, $LW_{OUT}$, Rn
and LE at daily scale, and reduced the RMSE by 4.62 W/m² to 14.64 W/m² and
increased the R² by 0.04 to 0.41 for $SW_{IN}$, $LW_{IN}$, $LW_{OUT}$, Rn, LE and H at 8-day scale,
when these estimates were validated against in situ observations at 134 EC sites.
Furthermore, the CoSEB-based datasets effectively captured the spatial-temporal
variability of global land surface radiation and heat fluxes, aligning well with those
from the mainstream products.
Our developed datasets hold significant potential for application across diverse
fields such as agriculture, forestry, hydrology, meteorology, ecology, and environmental
science. They can facilitate comprehensive studies on the variability, impacts, responses,
adaptation strategies, and mitigation measures of global and regional land surface
radiation and heat fluxes under the influences of climate change and human activities.
These datasets will provide valuable insights and data support for scientific research,
policy-making, and environmental management, advancing global solutions to address
climate change.

## Author contribution

JW: Writing – original draft, Visualization, Software, Formal analysis, Data
curation. RT: Writing – original draft, Validation, Supervision, Methodology, Funding
acquisition, Formal analysis, Conceptualization. ML: Writing – review & editing,
Validation. ZL: Writing – review & editing.

## Competing interests

The authors declare that they have no conflict of interest.

## Acknowledgment

We thank the work from the AmeriFlux, FLUXNET, EuroFlux, OzFlux,
ChinaFLUX, the National Tibetan Plateau/Third Pole Environment Data Center and
SURFRAD for providing in situ measurements. We would also like to thank Dr. Martin
Jung and Dr. Ulrich Weber for providing the FLUXCOM Bowen ratio-corrected
products. This work is supported by the National Natural Science Foundation of China
[42271378], and the Strategic Priority Research Program of the Chinese Academy of
Sciences (Grant No. XDB0740202).



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
