# Peer review of "Energy-conservation datasets of global land surface radiation"

_Earth System Science Data, 2025_

## Referee Comment (RC2)

**Review of Energy-conservation datasets of global land surface radiation and heat fluxes from 2000-2020 generated by CoSEB**

**Summary and recommendation-** In this paper , the authors apply a model of Coordinated estimates of land surface energy balance components (CoSEB) to generate estimates of surface radiation and heat fluxes from 2000 to 2020. An advantage of the CoSEB based approach is that estimates of radiation and heat are in "harmony" as opposed to generating independent estimates of each. The authors compare their estimates against observations from eddy covariance sites, other individual estimates and other individual observations. The paper is generally well written, and the results are presented clearly. However, I had several questions about the CoSEB framework itself and also the validations applied here in the manuscript. Hence I recommend major revisions. I have presented major comments and specific comments below.

**Major comments-**

1. **Explanation of updates to the CoSEB framework**- While reading the manuscript I realized that it is not only a paper that applies the existing CoSEB framework that is already published but also updates this framework to estimate to estimate radiation (previously this model estimated only land surface energy components and not short wave and long wave radiation). Therefore, authors need to discuss the effect of the addition of additional predicted variables on the equations and the results of the random forest. In particular, can the authors discuss which of the predictors were found to be the most important and also discuss how this differed with their previous publication? Also, can authors discuss generic details such as how many splits were generated by the random forest before and after the updates. Authors should also discuss the directionality of effects of different predictor variables based on the revised random forest.

2. **Multi-collinearity amongst predictor variables**- Authors should also discuss how multi-collinearity is handled amongst predictor variables given the large number of predictors. As far as I understand, random forests do not explicitly deal with multi collinearity unlike a PCA based approach for example. This can affect variable importance significantly. I would suggest authors explore this in detail.

3. **Effect of autocorrelation**- Given the temporal nature of several predictor variables, can authors confirm that autocorrelation does not exist or is minimized in their framework? What tests were performed to check for this? In particular I would recommend authors add lagged variables to the model to make sure that this is not the case. I believe several models constructed for earth system variables tend to ignore aspects such as autocorrelation and therefore this is an important point to address.

4. **Effect of downscaling ERA5- Land datasets**- The authors note on lines 195-197 that the ERA 5 land datasets used here have been downscaled from a resolution of ~9 kms to

~500m. This is a significant level of downscaling performed using a rather simple cubic convolution method. There are several variables related to the land cover (such as the LAI for example) that are used as predictor variables in the author's framework. Can the authors address the uncertainty caused by such large downscaling between scales on their results? On the one hand, based on the results, it seems that the model has produced reliable results compared to observations and other datasets even after such large downscaling. Is it that the land cover related variables do not play an important role in the predictions?

5. **In sample vs out of sample testing**- While the authors present significant comparisons with observations and other datasets to validate their model (e.g. Figure 3, Figure 4 and Figure 5), it seems the authors have not checked for overfitting of their approach by splitting the dataset into a training vs testing dataset. This is especially important since as mentioned in Major comment 1., the CoSEB framework itself has been updated. Authors should address this in detail. In fact, looking at Figure 3, it seems that the R squared values for G and H are on the lower side. I am curious as to what the values look like when out of sample testing is conducted?

Specific comments-

1. Abstract lines 31-36- The RMSEs presented here do not make any sense at this point since the reader has no sense of scale of values to expect. I recommend authors report the R squared values here instead. Also make sure to report whether the R squared is based on pooled data or just the testing data (See Major comment 5)

2. Introduction lines 74-75- Can the authors differentiate the citations between those for physical vs those for statistical methods.

3. Introduction line 92- "impending" is an awkward word here. I would just say "It was imperative".

4. Data lines 131-132- Why could a simple interpolation not be applied for missing half hourly data? Is the data extremely sensitive to time? Some clarification is needed here.

5. Data lines 138-139- Can the authors clarify why this criteria was applied for screening outliers?

6. Mainstream datasets/products for inter comparison- I was curious as to why the authors so not compare their estimates with heat and radiation estimates from popular earth system modelling systems such as CESM and CTSM (https://www.cesm.ucar.edu/). In fact, if the authors approach can produce estimates similar to earth system models, this would be a huge benefit to the community (since these models are laborious to run)

7. Methods lines 243-244- Once again the usage of RMSEs here does not make much sense. Can the authors just report the R squared values instead.

8.  Methods lines 269-270- Just to confirm, the RF based uncoordinated models are models where only individual variables are estimated rather than the simultaneous calculation of several variables? This should be clarified.

9.  Results Lines 306-309- I was curious looking at Figure 4 whether there were correlations or relationships between the EIR or RIR values and any of the other predictor variables? Is the shape of that distribution affected by any particular variables?

10.  Results Lines 311-312- Can the authors clarify the differences between site-based validation vs sample-based validation?

11.  Results lines 381-382- Once again, the RMSE values don't make a lot of sense here. Authors should report the R squared values instead.

12.  Section 4.2- When discussing the differences between the CoSEB model estimates vs other estimates, can authors also describe why the differences occur? A detailed discussion is not warranted here. Rather, I was interested in the author's perspective as to why the author's approach produces some differences over existing approaches.

---

## Author Comment (AC1)

Reviewer #2:

**Review of Energy-conservation datasets of global land surface radiation and heat fluxes from 2000-2020 generated by CoSEB**

**Summary and recommendation-** In this paper, the authors apply a model of Coordinated estimates of land surface energy balance components (CoSEB) to generate estimates of surface radiation and heat fluxes from 2000 to 2020. An advantage of the CoSEB based approach is that estimates of radiation and heat are in "harmony" as opposed to generating independent estimates of each. The authors compare their estimates against observations from eddy covariance sites, other individual estimates and other individual observations. The paper is generally well written, and the results are presented clearly. However, I had several questions about the CoSEB framework itself and also the validations applied here in the manuscript. Hence I recommend major revisions. I have presented major comments and specific comments below.

Ans: Thank you very much for your thoughtful and constructive comments. We sincerely appreciate your recognition of the CoSEB model and the datasets, particularly the advantage of generating global surface radiation and heat fluxes that adhere to energy conservation. We have carefully considered all the comments and suggestions from you and another reviewer, especially your concerns regarding the CoSEB framework and the validation of the datasets, and have made corresponding modifications and clarifications in the revised manuscript. More detailed information of our revisions can be found in the item-by-item response as below.

**Major comments-**
1. **Explanation of updates to the CoSEB framework-** While reading the manuscript I realized that it is not only a paper that applies the existing CoSEB framework that is already published but also updates this framework to estimate to estimate radiation (previously this model estimated only land surface energy components and not short wave and long wave radiation). Therefore, authors need to discuss the effect of the addition of additional predicted variables on the equations and the results of the random forest. In particular, can the authors discuss which of the predictors were found to be the most important and also discuss how this differed with their previous publication? Also, can authors discuss generic details such as how many splits were generated by the random forest before and after the updates. Authors should also discuss the directionality of effects of different predictor variables based on the revised random forest.

Ans: We thank the reviewer for these insightful comments and questions. Indeed, the renewed CoSEB model extends beyond the original version (Wang et al., 2025) by jointly estimating both radiation components ($SW_{IN}$, $SW_{OUT}$, $LW_{IN}$, $LW_{OUT}$ and Rn) and heat fluxes (LE, H, G), thereby ensuring that both radiation and energy balance are simultaneously satisfied.

(1) To illustrate the effect of including additional radiation components ($SW_{IN}$, $SW_{OUT}$, $LW_{IN}$ and $LW_{OUT}$) in the renewed CoSEB model compared with the original

version by Wang et al. (2025), we have tested the performance of a reconstructed model that estimated only Rn, LE, H and G using the same independent variables and samples as those in the renewed CoSEB model. The results (Fig. S5 in the supplementary material) showed no significant differences from those produced by the renewed CoSEB model, indicating that the expansion of radiation components did not compromise the model's overall performance. We have discussed this in the second paragraph of Section 5 with the following sentences:

"Furthermore, to better illustrate the effect of including additional radiation components ($SW_{IN}$, $SW_{OUT}$, $LW_{IN}$ and $LW_{OUT}$) in the renewed CoSEB model compared with the original version by Wang et al. (2025), we have tested the performance of a reconstructed model that estimated only Rn, LE, H and G using the same independent variables and samples as those in the renewed CoSEB model. The results (Fig. S5 in the supplementary material) showed no significant differences in accuracy compared with those of the renewed CoSEB model, indicating the expansion of radiation components did not compromise model performance."

[Figure]

**Fig. S5 Scatter density plots of the site-based 10-fold cross-validation of daily net radiation (Rn), soil heat flux (G), latent heat flux (LE) and sensible heat flux (H) derived by a reconstructed model within the CoSEB framework against in-situ observed Rn, G, and energy imbalance-corrected LE ( $LE_{daily}^{corr}$ ) and H ( $H_{daily}^{corr}$ ), where the model was designed to estimate only four of the eight flux components. The EIR in the subfigure (e) represents the energy imbalance ratio, which are defined as 100% × (Rn - G - LE - H)/Rn. The colorbar represents the normalized density of data points.**

(2) Regarding your concern about the importance of the feature variables to the renewed CoSEB model, we have added a new table (Table S4 in the Supplementary Material) to show the importance scores of different feature variables using the builtin method of the random forests. The results showed that solar radiation reaching the surface of the earth is the most important variable, which is consistent with the results from our previous study (Wang et al., 2025). We have discussed this in the second paragraph of Section 5 with the following sentences:

"The importance scores of the 19 different feature variables are exhibited in Table S4 in the Supplementary Material, and downward solar radiation, the primary source of the energy at the earth surface, is the most important input variable, consistent with the results from our previous study (Wang et al., 2025)."

Table S4 Importance scores of the 19 different feature variables in the construction of the renewed CoSEB model for estimating daily downward shortwave and longwave radiation ($SW_{IN}$ and $LW_{IN}$), upward shortwave and longwave radiation ($SW_{OUT}$ and $LW_{OUT}$), net radiation (Rn), latent heat flux (LE), sensible heat flux (H) and soil heat flux (G).

| Types | Features Variables | Abbreviation | Importance Score | Cumulative Percentage (%) |
|---|---|---|---|---|
| Climate/meteorology | solar radiation reaching the surface of the earth | $SW_{IN}^{ERA5}$ | 0.5724 | 57.24 |
| Climate/meteorology | 2 m air temperature | $T_a$ | 0.2338 | 80.62 |
| Vegetation and soil | Fractional tree cover | FVC | 0.0292 | 83.54 |
| Climate/meteorology | net thermal radiation at the surface | $LW_{net}$ | 0.0241 | 85.95 |
| Vegetation and soil | Leaf area index | LAI | 0.0241 | 88.36 |
| Vegetation and soil | Percent tree cover | PTC | 0.0177 | 90.13 |
| Vegetation and soil | soil temperature in surface layer | $T_{SI}$ | 0.0107 | 91.20 |
| Climate/meteorology | surface air pressure | PA | 0.0097 | 92.17 |
| Topography | Surface slope | Slope | 0.0093 | 93.10 |
| Climate/meteorology | precipitation | $P_r$ | 0.0091 | 94.01 |
| Others | inverse relative distance from the Earth to the Sun | dr | 0.0089 | 94.9 |
| Others | latitude | Lat | 0.0075 | 96.65 |
| Climate/meteorology | Relative air humidity | RH | 0.0074 | 96.39 |
| Topography | Digital elevation model | DEM | 0.0072 | 97.11 |
| Vegetation and soil | soil volumetric moisture content in surface layer | SM1 | 0.007 | 97.81 |
| Others | longitude | Lon | 0.0067 | 98.48 |
| Climate/meteorology | Carbon dioxide concentration | $CO_2$ | 0.0056 | 99.04 |
| Topography | Surface aspect | Aspect | 0.005 | 99.54 |
| Climate/meteorology | Wind speed | WS | 0.0046 | 100 |

(3) We have added a brief description of the optimization of hyperparameters for the renewed CoSEB model using the random search method and grid search method. Specifically, the number of decision trees, the max depth, min samples split, and min samples leaf of the MRF are set to 281, 21, 8, and 8, respectively, compared to 295, 20, 12, and 8 in our previous study of Wang et al. (2025). The corresponding details have been added at the beginning of the third paragraph of Section 3 in the revised manuscript with the following sentences:

"To enhance model generalization, the renewed CoSEB model was reoptimized using random and grid search methods, resulting in different hyperparameters of 281 decision trees, a maximum depth of 21, and minimum samples split and leaf of 8 from those of Wang et al. (2025)."

(4) We would like to emphasize that the main focus of this study was to develop the data-driven energy-conservation global datasets using multiple input variables that have certain influences on surface radiation and heat fluxes, rather than to explore the directionality of effects of each input variable on surface radiation and heat fluxes. Since directionality analysis does not alter model parameters, affect

model construction, or impact the generation of the CoSEB-based datasets, in almost no articles (Jung et al., 2019; Mu et al., 2011; Ryu et al., 2018; Xu et al., 2022) focusing on models and algorithms for surface radiation fluxes and heat fluxes have we seen anyone conduct directionality analysis; therefore, conducting directionality analysis is not necessary within the scope of our study.

2. Multi-collinearity amongst predictor variables- Authors should also discuss how multi-collinearity is handled amongst predictor variables given the large number of predictors. As far as I understand, random forests do not explicitly deal with multi collinearity unlike a PCA based approach for example. This can affect variable importance significantly. I would suggest authors explore this in detail.

Ans: We thank the reviewer for this comment. While random forests do not explicitly eliminate multi-collinearity among input variables, they randomly select subsets of input features at each split (Breiman, 2001) and are generally considered robust in terms of performance even when multi-collinearity exists among some inputs (Drobnič et al., 2020). Besides, in selecting the input variables, prior knowledge derived from previous studies was employed to identify factors that exert significant influence on surface radiation and heat flux while maintaining relative inter-independence. This practice is widely adopted in data-driven models for estimating land surface water, energy, and carbon fluxes (Bai et al., 2024; Elghawi et al., 2023; Han et al., 2023; O. & Orth, 2021), and few studies specifically perform multicollinearity analysis before modeling. Although some of the selected variables may exhibit a certain degree of multi-collinearity, each carries unique characteristic information, making it inappropriate to consider only a single dominant variable during model construction. Moreover, we acknowledge that variable importance should be interpreted with caution, since the importances may not be accurate in the presence of multicollinearity. However, we would also like to clarify that the primary aim of this study was to improve the accuracy of the developed datasets rather than to interpret the individual contributions of each input variable. We have discussed this in second paragraph of Section 5 with the following sentence:

"In selecting the 19 input variables to accommodate the additional target variables, prior knowledge derived from previous studies was employed to identify factors that exert significant influence on surface radiation and heat flux while maintaining relative inter-independence as much as possible (Jung et al., 2019; Mohan et al., 2020; Wang et al., 2021; Xian et al., 2024). This practice is commonly adopted in data-driven models for estimating land surface water, energy, and carbon fluxes (Bai et al., 2024; Elghawi et al., 2023; Han et al., 2023; O. & Orth, 2021). The importance scores of the 19 different feature variables are exhibited in Table S4 in the Supplementary Material, and downward solar radiation, the primary source of the energy at the earth surface, is the most important input variable, consistent with the results from our previous study (Wang et al., 2025). Although some of the selected variables may exhibit a certain degree of multi-collinearity, each contributes unique and physically meaningful information, supporting the inclusion of all variables in model construction. Note that the variable importance, derived from the built-in

method of the random forests and potentially affected by multicollinearity among the input variables, is presented only as a reference. Retaining all 19 feature variables ensures the model's flexibility and generalization capability, enabling future incorporation of additional representative ground-based observations for further training and improvement."

3. Effect of autocorrelation- Given the temporal nature of several predictor variables, can authors confirm that autocorrelation does not exist or is minimized in their framework? What tests were performed to check for this? In particular I would recommend authors add lagged variables to the model to make sure that this is not the case. I believe several models constructed for earth system variables tend to ignore aspects such as autocorrelation and therefore this is an important point to address.

Ans: Thanks for your question and suggestion. We agree that several predictor variables may exhibit autocorrelation. To investigate the impact of lagged effects of input variables on model performance, we specifically conducted an experiment by including lagged air temperature (i.e., the air temperature of the previous day, because air temperature, identified alongside downward solar radiation as one of the two most influential variables in the model based on the importance scores in Supplementary Table S4, exhibits a more pronounced lagged effect than solar radiation) as additional predictor. The results (Fig. S4 in the Supplementary Material) showed no noticeable improvement in model accuracy, suggesting that lagged effects were negligible in the CoSEB framework for estimates of daily surface radiation and heat fluxes. We speculate that lagged effects may have a more pronounced influence on flux estimates at higher temporal resolutions (e.g., half-hourly), but this is beyond the scope of the present study. We have discussed this in the second paragraph of Section 5 with the following sentence:

"Besides, to investigate the impact of lagged effects of input variables on model performance, experiments were also conducted by adding lagged variables (e.g., the air temperature of the previous day) to the 19 input features. The results (Fig. S4 in the Supplementary Material) showed almost no improvement in model accuracy, suggesting that lagged effects on model performance were negligible within the CoSEB framework for estimates of daily surface radiation and heat fluxes."

[Figure]

**Fig. S4** Scatter density plots of the site-based 10-fold cross-validation of daily downward shortwave and longwave radiation ($SW_{IN}$ and $LW_{IN}$), upward shortwave and longwave radiation ($SW_{OUT}$ and $LW_{OUT}$), net radiation (Rn), soil heat flux (G), latent heat flux (LE) and sensible heat flux (H) derived by a reconstructed model within the CoSEB framework against in situ observed $SW_{IN}$, $LW_{IN}$, $SW_{OUT}$, $LW_{OUT}$, Rn, G, and energy imbalance-corrected LE ( $LE_{daily}^{corr}$ ) and H ($H_{daily}^{corr}$), where the air temperature of the previous day was additionally added to the 19 input feature variables of the model as the lagged variable. The EIR and RIR in the subfigure (i) represent the energy imbalance ratio and radiation imbalance ratio, which are defined as 100% × (Rn - G - LE - H)/Rn and 100% × ($SW_{IN}$ − $SW_{OUT}$ + $LW_{IN}$ - $LW_{OUT}$)/Rn, respectively. The colorbar represents the normalized density of data points.

4. Effect of downscaling ERA5- Land datasets- The authors note on lines 195-197 that the ERA 5 land datasets used here have been downscaled from a resolution of ~9 kms to ~500m. This is a significant level of downscaling performed using a rather simple cubic convolution method. There are several variables related to the land cover (such as the LAI for example) that are used as predictor variables in the author's framework. Can the authors address the uncertainty caused by

such large downscaling between scales on their results? On the one hand, based on the results, it seems that the model has produced reliable results compared to observations and other datasets even after such large downscaling. Is it that the land cover related variables do not play an important role in the predictions?

Ans: Thanks for your comment and question. We would like to clarify that the ERA5-Land datasets used in this study mainly include meteorological reanalysis variables (e.g., solar radiation, pressure of the atmosphere, wind speed and relative air humidity), which were downscaled from their original ~9 km spatial resolution to 500 m. In contrast, the land cover-related vegetation variables, including LAI, FVC, and PTC, were directly obtained from remote sensing products such as MODIS and GLASS (see Section 2.2), which already have an original spatial resolution of ~500 m and therefore did not require spatial downscaling.

Besides, we acknowledge that downscaling ERA5-Land datasets from ~9 km to ~500 m using a cubic convolution method may introduce certain uncertainties. However, this resampling was necessary to match the footprint of the site-based measurements of turbulent heat fluxes, which is a common practice in the generation of remote sensing products (Mu et al., 2011; Ryu et al., 2018; Senay et al., 2020; Zhang et al., 2019; Zheng et al., 2022). Moreover, the machine learning framework of the CoSEB model can partially mitigate such uncertainties introduced by the downscaling during training by learning complex relationships among multiple inputs and in situ observed energy components. This is reflected in the good agreement of the CoSEB-based estimates with both in-situ observations and other mainstream products. Our previous studies (Wang et al., 2025, the last paragraph of Section 5.1) also have demonstrated that the differences in meteorological reanalysis data caused by spatial downscaling have a relatively small impact on the estimates by the machine-learning-based CoSEB model.

Furthermore, it is also important to note that this does not imply that land-cover-related variables do not play an important role in the estimations. As shown by the variable importance scores presented in the newly added Table S4 in the Supplementary Material, vegetation and surface-related parameters such as FVC and LAI have high importance scores. These variables can partially compensate for the spatial heterogeneity and localized variations not captured by the coarse-resolution ERA5-Land datasets, thereby enhancing the performance of the model.

We have discussed this in the last paragraph of Section 5 with the following sentence:

"Another potential source of uncertainty arises from differences in meteorological reanalysis data caused by spatial downscaling, which, as demonstrated in our previous study (Wang et al., 2025, the last paragraph of Section 5.1), has a relatively small impact on model estimates by the machine-learning-based CoSEB model combined with finer-resolution surface-related variables that partially compensate for the spatial heterogeneity and localized variations not captured by the coarse-resolution datasets."

5. In sample vs out of sample testing- While the authors present significant

comparisons with observations and other datasets to validate their model (e.g. Figure 3, Figure 4 and Figure 5), it seems the authors have not checked for overfitting of their approach by splitting the dataset into a training vs testing dataset. This is especially important since as mentioned in Major comment 1., the CoSEB framework itself has been updated. Authors should address this in detail. In fact, looking at Figure 3, it seems that the R squared values for G and H are on the lower side. I am curious as to what the values look like when out of sample testing is conducted?

Ans: We appreciate the reviewer's insightful comments and questions. We would like to clarify that the out-of-sample testing of the updated CoSEB model has already been evaluated using site-based 10-fold cross-validation. In this approach, all sites were divided into ten folds, where the samples from each fold of sites in turn served as validation datasets while the remaining folds were used for training. This ensures that the validation datasets are spatially independent from the training datasets, effectively serving as out-of-sample testing. The results shown in Figure 3, corresponding to the site-based 10-fold cross-validation, showed that the $R^2$ values for H and G are 0.59 and 0.42, respectively. We have already described the site-based 10-fold cross-validation in the third paragraph of Section 3 with the following sentence:

"Site-based 10-fold cross-validation was employed to evaluate the transferability and generalization of the CoSEB model by randomly dividing all sites into ten folds, where the samples from each fold of sites in turn served as validation datasets while the remaining folds were used as training datasets, ensuring that the validation was conducted on sites spatially independent from the training data."

Furthermore, to evaluate potential overfitting, the mean RMSE and $R^2$ values along with their standard deviations across the ten folds of the site-based cross-validation have been presented in Table S3 of the Supplementary Material. Comparisons between the training results (Table S3) and validation results (Fig. 3) indicate that, although the CoSEB model performs better on the training datasets than on the validation datasets, the overall performance remains stable. This stability, particularly given that the validation is conducted on spatially independent sites, demonstrates that the model is not affected by overfitting. We have illustrated this in the first paragraph of Section 4.1.1 with the following sentence:

"Comparisons with the corresponding training results (Table S3 in the Supplementary Material) indicated that although the CoSEB model performed better on the training datasets, its overall performance remained stable, suggesting that the CoSEB model was not affected by overfitting."

Table S3 The mean root mean square error (RMSE) and coefficient of determination ($R^2$) along with their standard deviations across the ten folds of the site-based cross-validation for the renewed CoSEB model.

|  | RMSE (W/m$^2$) | $R^2$ |
|---|---|---|
| $SW_{IN}$ | 28.56±0.09 | 0.91±0.001 |
| $SW_{OUT}$ | 9.83±0.10 | 0.79±0.003 |
| $LW_{IN}$ | 12.41±0.08 | 0.95±0.001 |
| $LW_{OUT}$ | 8.52±0.07 | 0.97±0.001 |
| Rn | 22.49±0.08 | 0.85±0.001 |
| LE | 19.75±0.15 | 0.82±0.003 |
| H | 19.36±0.12 | 0.76±0.003 |
| G | 5.39±0.04 | 0.60±0.004 |

Specific comments-
1. Abstract lines 31-36- The RMSEs presented here do not make any sense at this point since the reader has no sense of scale of values to expect. I recommend authors report the R squared values here instead. Also make sure to report whether the R squared is based on pooled data or just the testing data (See Major comment 5)

Ans: We appreciate the reviewer's constructive suggestion. We would like to clarify that RMSE remains a key metric for evaluating the accuracy of the model and datasets, particularly for energy flux estimations (Bisht & Bras, 2011; Comini De Andrade et al., 2024; Kalma et al., 2008; Ryu et al., 2008; Zhang et al., 2019), as it directly quantifies prediction errors in physical units (W/m$^2$), making it an indicator of significant interest to both model developers and product users. However, $R^2$ indeed is another important metric, indicating the degree to which the model predictions align with the reference truth. Therefore, in the revised Abstract, we have reported both RMSE and $R^2$ values for the CoSEB-based datasets. In addition, we have clarified that the reported RMSE and $R^2$ values of the CoSEB-based datasets are derived from validation at independent test datasets across 44 sites (see Section 2.1). The revised sentences are as follows:

"(1) the RMSEs ($R^2$) for daily estimates of $SW_{IN}$, $SW_{OUT}$, $LW_{IN}$, $LW_{OUT}$, Rn, LE, H and G from the CoSEB-based datasets at 44 independent test sites were 37.52 W/m$^2$ (0.81), 14.20 W/m$^2$ (0.42), 22.47 W/m$^2$ (0.90), 13.78 W/m$^2$ (0.95), 29.66 W/m$^2$ (0.77), 30.87 W/m$^2$ (0.60), 29.75 W/m$^2$ (0.44) and 5.69 W/m$^2$ (0.44), respectively,"

2. Introduction lines 74-75- Can the authors differentiate the citations between those for physical vs those for statistical methods.

Ans: Thanks for your valuable suggestion. We have clearly differentiated the citations between those for physical vs those for statistical methods in the revised manuscript as follows:

"In past decades, numerous RS-based products/datasets of global surface radiation and heat fluxes have significantly advanced, which were generally generated by physical (Li et al., 2023; Mu et al., 2011; Yu et al., 2022) or statistical methods (Jiao et al., 2023; Jung et al., 2019; Peng et al., 2020)."

3. Introduction line 92- "impending" is an awkward word here. I would just say "It was imperative".

Ans: We appreciate the reviewer's suggestion. We have revised this sentence to "It was imperative to develop global datasets of land surface radiation and heat fluxes characterized by high accuracies, radiation balance as well as heat balance, to better meet the requirements in practical applications of various fields." in the new manuscript.

4. Data lines 131-132- Why could a simple interpolation not be applied for missing half hourly data? Is the data extremely sensitive to time? Some clarification is needed here.

Ans: Thank you for your comments and questions. The half-hourly surface radiation and heat fluxes are sensitive to short-term temporal variations caused by rapid changes in meteorological conditions, but their intraday dynamics are often nonlinear, particularly due to the intermittent effects of cloud cover. Therefore, applying simple interpolation methods (e.g. linear interpolation) could introduce considerable uncertainties. To ensure data quality, we only retained directly observed values (data quality flag=0) and good-quality gap-filled data (data quality flag=1) provided by the official gap-filling algorithms, and then computed daily averages only when more than 80% of half-hourly observations were available, as already described in the first paragraph of Section 2.1 with the following sentence:

"(3) the half-hourly ground-based observations with quality-control flag of 2 or 3 (bad quality) were removed but quality-control flag of 0 and 1 (good quality) were maintained; (4) a daily average of the half-hour observations was calculated for each day with greater than 80% good-quality data, further reducing the 472 sites to 355 sites;"

Besides, we have already discussed the uncertainties caused by the daily averages of surface radiation and heat fluxes in the last paragraph of Section 5 with the following sentence:

"Specifically, daily averages of surface radiation and heat fluxes for each day were obtained for analysis from good-quality half-hourly observations when the fraction of these good-quality half-hourly observations was greater than 80% in a day, due to the lack of consensus on the method for aggregating gapped half-hourly observations to daily data (Tang et al., 2024a; Yao et al., 2017; Zheng et al., 2022)."

Following your suggestion, we have also further clarified the simple temporal interpolation in the last paragraph of Section 5 with the following sentence:

"Simple temporal interpolation of half-hourly in situ observations, which could therefore introduce substantial uncertainties, was not applied, because surface radiation and heat fluxes are sensitive to short-term variations in meteorological conditions and their intraday dynamics are often complex."

5. Data lines 138-139- Can the authors clarify why this criteria was applied for screening outliers?

Ans: Thank you for your valuable question. We would like to clarify that the energy balance ratio (EBR) of 0.2-1.8 and the 1st-99th quantiles of the daily evaporation fraction was both applied to remove physically implausible measurements, such as cases where the available surface energy (Rn − G) is close to zero while LE and H remain comparatively large, where the threshold of 0.2-1.8 was adopted following our previous study (Wang et al., 2025), which has demonstrated that nearly all available data fall within this range and that the accuracy of the CoSEB model showed no significant differences when applying different EBR thresholds, while the percentile-based screening was employed following common practice in flux and remote sensing studies (Bartkowiak et al., 2024; Ghorbanpour et al., 2022; Wang et al., 2023). We have clarified this in the first paragraph of Section 2.1 with the following sentence:

"(5) the aggregated daily LE and H were corrected for energy imbalance using the Bowen ratio method when the daily energy balance closure [defined as $(LE+H)/(Rn-G)$] varied between 0.2 and 1.8 following Wang et al. (2025) to exclude physically implausible measurements; (6) extreme outliers in the daily evaporative fraction were further removed by excluding values outside the 1st–99th percentile range, a common practice in flux and remote sensing studies (Bartkowiak et al., 2024; Wang et al., 2023), further reducing the 355 sites to 337 sites."

6. Mainstream datasets/products for inter comparison- I was curious as to why the authors so not compare their estimates with heat and radiation estimates from popular earth system modelling systems such as CESM and CTSM (https://www.cesm.ucar.edu/). In fact, if the authors approach can produce estimates similar to earth system models, this would be a huge benefit to the community (since these models are laborious to run)

Ans: Thanks for your comment. The outputs of Earth system models generally have coarse spatial resolutions (e.g., the CESM Large Ensemble Project has a spatial resolution of ~1°). Due to the surface heterogeneity, these model outputs cannot be directly validated using radiation and heat flux observations from ground sites with limited spatial representativeness. This is the main reason why both we and others usually do not compare the outputs of Earth system models with remote sensing-based datasets.

Although we believe that comparing the outputs of Earth system models with remote sensing-based datasets (including our CoSEB-based datasets and others' PML_V2, MOD16A2, FLUXCOM, BESSV2.0, GLASS) and validating them against ground-based observations is not appropriate, following the reviewer's suggestion, we compared the global spatial distributions of mean annual estimates from CoSEB-based datasets with the outputs from the CESM Large Ensemble project. The results (see Section 4.3 and Fig. S8) show that, overall, the global spatial patterns of the estimated $SW_{IN}$, $LW_{IN}$, $LW_{OUT}$, Rn, LE and H are consistent,

though numerical differences exist. Considering the scope and length of the current manuscript, a more detailed analysis of the spatial-temporal distribution patterns, trends, and variability between Earth system model outputs and remote sensing-based datasets could be conducted in future work. We have discussed this in the third paragraph of Section 5 with the following sentences:

"Preliminary analysis indicates that the CoSEB-based datasets exhibit spatial patterns consistent with those of mainstream RS-based datasets and Earth system model outputs (see Fig. S8 in the supplementary material). More detailed analysis about their similarities and differences can be further conducted in future work."

[Figure]

**Fig. S8 Spatial patterns of global mean annual downward shortwave radiation ($SW_{IN}$), downward longwave radiation ($LW_{IN}$), upward longwave radiation ($LW_{OUT}$), net radiation (Rn), latent heat flux (LE) and sensible heat flux from 2001 to 2018 by Community Earth System Model (CESM) Large Ensemble project, where $LW_{OUT}$ and Rn were inferred from surface radiation balance and heat balance.**

7. Methods lines 243-244- Once again the usage of RMSEs here does not make much sense. Can the authors just report the R squared values instead.

Ans: We appreciate the reviewer's suggestion. We would like to clarify that RMSE remains an essential metric for evaluating the accuracy of the model and datasets, particularly for energy flux estimations (Bisht & Bras, 2011; Comini De Andrade et al., 2024; Kalma et al., 2008; Ryu et al., 2008; Zhang et al., 2019), as it directly quantifies prediction errors in physical units ($W/m^2$), making it an indicator of significant interest to both model developers and product users. Nevertheless, $R^2$ indeed is another important metric, indicating the degree to which the model predictions align with the reference truth. After careful consideration, we have additionally reported $R^2$ values in the revised manuscript to more comprehensively demonstrate the model performance. The revised sentence is as follows:

"The CoSEB model was demonstrated to be able to produce high-accuracy estimates of land surface energy components, with the RMSE of <17 $W/m^2$ and $R^2$ of > 0.83 for estimating 4-day Rn, LE and H, and the RMSE of <5 $W/m^2$ and $R^2$ of 0.55 for estimating 4-day G."

8. Methods lines 269-270- Just to confirm, the RF based uncoordinated models are models where only individual variables are estimated rather than the simultaneous calculation of several variables? This should be clarified.

Ans: Thanks for your valuable question. Your understanding is correct. We have more clearly clarified this in the third paragraph of Section 3 of the revised manuscript with the following sentence:

"Furthermore, to benchmark the coordinated estimates from the renewed CoSEB model, eight RF-based uncoordinated models were constructed, each separately estimating one of $SW_{IN}$, $SW_{OUT}$, $LW_{IN}$, $LW_{OUT}$, Rn, LE, H or G using the same inputs as those in the renewed CoSEB model."

9. Results Lines 306-309- I was curious looking at Figure 4 whether there were correlations or relationships between the EIR or RIR values and any of the other predictor variables? Is the shape of that distribution affected by any particular variables?

Ans: Thanks for your question. We would like to clarify that our CoSEB model showed no energy imbalance, with the RIR and EIR of 0, as shown in Figure 3. The distributions of RIR and EIR in Figure 4 were derived from RF-based uncoordinated models, which were used only for comparison with our CoSEB model and were not the focus of our study.

However, considering your concern about whether the distributions of the RIR and EIR values are affected by specific predictor variables, we further conducted a binned statistical analysis, where the three most critical input variables identified in Table S4 (i.e. $SW_{IN}^{ERA5}$, $T_a$ and FVC) were divided into equal-width bins, and for each bin the mean and standard deviation for positive and negative RIR conditions were calculated. Besides, the Pearson correlation coefficients (r) between RIR (EIR) and each input variable were computed to quantify their overall relationships. The results showed that lower levels of solar radiation, air temperature, or FVC are associated with larger RIR (EIR), while the predominance of low values of these three variables tends to result in decreased kurtosis correspondingly, implying flatter and broader probability shapes of RIR and EIR. We have also briefly illustrated this in the end of the second paragraph of Section 4.1.1 with the following sentence:

"Furthermore, the RIR as well as EIR tended to be higher under lower solar radiation, air temperature, or FVC, with more frequent low values of these three variables leading to a broader and less peaked distribution of RIR and EIR (see Fig. S1 in the Supplementary Material)."

[Figure]

**Fig. S1 Relationships between radiation imbalance ratio [RIR, 100% × ($SW_{IN} - SW_{OUT} + LW_{IN} - LW_{OUT}$)/Rn] and energy imbalance ratio [EIR, 100% × (Rn - G - LE - H)/Rn] derived from RF-based uncoordinated models and three critical input variables identified in Table S4, including solar radiation reaching the surface of the earth from ERA5-Land ($SW_{IN}^{ERA5}$, the first column), 2 m air temperature from ERA5-Land ($T_a$, the second column) and fraction vegetation cover from GLASS (*FVC*, the third column). The mean and standard deviation were calculated within equal-width bins of $SW_{IN}^{ERA5}$, $T_a$, and FVC under positive and negative EIR (RIR) conditions, where the solid lines represent the mean values, and the shaded area represents the corresponding variation of standard deviations. The r values in legends indicate the Pearson correlation coefficients.**

10. Results Lines 311-312- Can the authors clarify the differences between site-based validation vs sample-based validation?

Ans: We appreciate the reviewer's insightful comment. Sample-based 10-fold cross-validation refers to randomly splitting all available samples from all sites into ten folds, with each fold in turn serving as the validation dataset while the remaining folds are used for training. This approach allows samples from the same site to appear in both the training and validation datasets. In contrast, site-based 10-fold cross-validation was performed by randomly dividing all sites into ten folds, with the samples from each fold of sites used for validation in turn. This strategy ensures that the validation datasets are spatially independent from the training datasets, thereby providing a more rigorous assessment of the model's spatial generalization capability. We have already described the site-based 10-fold cross-validation in the third paragraph of Section 3 with the following sentences:

"Site-based 10-fold cross-validation was employed to evaluate the transferability and

generalization of the CoSEB model by randomly dividing all sites into ten folds, where the samples from each fold of sites in turn served as validation datasets while the remaining folds were used as training datasets, ensuring that the validation was conducted on sites spatially independent from the training data."

Furthermore, after careful consideration, site-based 10-fold cross-validation was deemed to be more suitable for assessing the performance of the model than sample-based 10-fold cross-validation, as the validation datasets in site-based cross-validation are spatially independent from the training datasets. To make the main focus of the manuscript clearer and more concise, we retained only the site-based 10-fold cross-validation and removed the sample-based 10-fold cross-validation in the revised manuscript.

11. Results lines 381-382- Once again, the RMSE values don't make a lot of sense here. Authors should report the R squared values instead.

Ans: We appreciate the reviewer's suggestion. We would like to clarify that RMSE remains an essential metric for evaluating the accuracy of the model and datasets, particularly for energy flux estimations (Bisht & Bras, 2011; Comini De Andrade et al., 2024; Kalma et al., 2008; Ryu et al., 2008; Zhang et al., 2019), as it directly quantifies prediction errors in physical units ($W/m^2$), making it an indicator of significant interest to both model developers and product users. However, $R^2$ indeed is another important metric, indicating the degree to which the model predictions align with the reference truth. After careful consideration, we have additionally incorporated the $R^2$ values into the revised manuscript. The revised sentence is as follows:

"Results indicated that the CoSEB-based datasets could provide good estimates of $SW_{OUT}$, H and G, with the RMSEs ($R^2$) of 14.20 $W/m^2$ (0.42), 29.75 $W/m^2$ (0.44) and 5.69 $W/m^2$ (0.44) at daily scale, respectively, and the RMSE ($R^2$) of 12.19 $W/m^2$ (0.39) and 4.60 $W/m^2$ (0.47) for 8-day $SW_{OUT}$ and G, respectively."

12. Section 4.2- When discussing the differences between the CoSEB model estimates vs other estimates, can authors also describe why the differences occur? A detailed discussion is not warranted here. Rather, I was interested in the author's perspective as to why the author's approach produces some differences over existing approaches.

Ans: Thanks for your constructive comments. The possible reasons for the differences between estimates from the CoSEB-based datasets and the mainstream products/datasets are complex and may arise from differences in both methodological frameworks and input datasets. Specifically, the discrepancies may result from the simplification of physical processes and the uncertainties in parameterization within the physics-based products (e.g., MOD16A1, BESSV2.0, PML_V2, and ETMonitor). In contrast, the differences between the CoSEB-based datasets and other machine-learning-based products (e.g., BESS-Rad, GLASS, and FLUXCOM) may be attributed to the limited sample sizes of training data, the

incomplete consideration of influencing factors (e.g., $CO_2$ concentration, surface aspect), and the lack of physical constraints among energy balance components in existing machine-learning frameworks. We have briefly discussed this in the last paragraph of Section 4.2 of the revised manuscript with the following sentence:

"The differences between the estimates from the CoSEB-based datasets and mainstream datasets are likely multifactorial, arising from the simplification and parameterization uncertainties in physics-based models, as well as the lack of physical constraints, limited training samples, and incomplete consideration of influencing factors in other machine-learning-based models."

Reference:

Bai, Y., Mallick, K., Hu, T., Zhang, S., Yang, S. and Ahmadi, A.: Integrating machine learning with thermal-driven analytical energy balance model improved terrestrial evapotranspiration estimation through enhanced surface conductance, Remote Sens. Environ., 311, 114308. 10.1016/j.rse.2024.114308, 2024.

Bartkowiak, P., Ventura, B., Jacob, A. and Castelli, M.: A Copernicus-based evapotranspiration dataset at 100 m spatial resolution over four Mediterranean basins, Earth Syst. Sci. Data, 16, 4709-4734. 10.5194/essd-16-4709-2024, 2024.

Bisht, G. and Bras, R. L.: Estimation of Net Radiation From the Moderate Resolution Imaging Spectroradiometer Over the Continental United States, IEEE Trans. Geosci. Remote Sensing, 49, 2448-2462. 10.1109/tgrs.2010.2096227, 2011.

BREIMAN, L.: Random forests, Mach. Learn., 45, 5-32. 2001.

Comini de Andrade, B., Laipelt, L., Fleischmann, A., Huntington, J., Morton, C., Melton, F., Erickson, T., Roberti, D. R., de Arruda Souza, V., Biudes, M., Gomes Machado, N., Antonio Costa dos Santos, C., Cosio, E. G. and Ruhoff, A.: geeSEBAL-MODIS: Continental-scale evapotranspiration based on the surface energy balance for South America, ISPRS-J. Photogramm. Remote Sens., 207, 141-163. 10.1016/j.isprsjprs.2023.12.001, 2024.

Drobnič, F., Kos, A. and Pustišek, M.: On the Interpretability of Machine Learning Models and Experimental Feature Selection in Case of Multicollinear Data, Electronics, 9. 10.3390/electronics9050761, 2020.

ElGhawi, R., Kraft, B., Reimers, C., Reichstein, M., Körner, M., Gentine, P. and Winkler, A. J.: Hybrid modeling of evapotranspiration: inferring stomatal and aerodynamic resistances using combined physics-based and machine learning, Environ. Res. Lett., 18, 034039. 10.1088/1748-9326/acbbe0, 2023.

Ghorbanpour, A. K., Kisekka, I., Afshar, A., Hessels, T., Taraghi, M., Hessari, B., Tourian, M. J. and Duan, Z.: Crop Water Productivity Mapping and Benchmarking Using Remote Sensing and Google Earth Engine Cloud Computing, Remote Sens., 14. 10.3390/rs14194934, 2022.

Han, Q., Zeng, Y., Zhang, L., Wang, C., Prikaziuk, E., Niu, Z. and Su, B.: Global long term daily 1 km surface soil moisture dataset with physics informed machine learning, Sci. Data, 10, 101. 10.1038/s41597-023-02011-7, 2023.

Jung, M., Koirala, S., Weber, U., Ichii, K., Gans, F., Camps-Valls, G., Papale, D., Schwalm, C.,

Tramontana, G. and Reichstein, M.: The FLUXCOM ensemble of global land-atmosphere energy fluxes, Sci. Data, 6, 74. 10.1038/s41597-019-0076-8, 2019.

Kalma, J. D., McVicar, T. R. and McCabe, M. F.: Estimating Land Surface Evaporation: A Review of Methods Using Remotely Sensed Surface Temperature Data, Surveys in Geophysics, 29, 421-469. 10.1007/s10712-008-9037-z, 2008.

Mu, Q., Zhao, M. and Running, S. W.: Improvements to a MODIS global terrestrial evapotranspiration algorithm, Remote Sens. Environ., 115, 1781-1800. 10.1016/j.rse.2011.02.019, 2011.

O., S. and Orth, R.: Global soil moisture data derived through machine learning trained with in-situ measurements, Sci. Data, 8. 10.1038/s41597-021-00964-1, 2021.

Ryu, Y., Jiang, C., Kobayashi, H. and Detto, M.: MODIS-derived global land products of shortwave radiation and diffuse and total photosynthetically active radiation at 5 km resolution from 2000, Remote Sens. Environ., 204, 812-825. 10.1016/j.rse.2017.09.021, 2018.

Ryu, Y., Kang, S., Moon, S.-K. and Kim, J.: Evaluation of land surface radiation balance derived from moderate resolution imaging spectroradiometer (MODIS) over complex terrain and heterogeneous landscape on clear sky days, Agric. For. Meteorol., 148, 1538-1552. 10.1016/j.agrformet.2008.05.008, 2008.

Senay, G. B., Kagone, S. and Velpuri, N. M.: Operational Global Actual Evapotranspiration: Development, Evaluation and Dissemination, Sensors (Basel), 20. 10.3390/s20071915, 2020.

Wang, J., Tang, R., Liu, M., Jiang, Y., Huang, L. and Li, Z.-L.: Coordinated estimates of 4-day 500 m global land surface energy balance components, Remote Sens. Environ., 326, 114795. 10.1016/j.rse.2025.114795, 2025.

Wang, Y., Hu, J., Li, R., Song, B. and Hailemariam, M.: Remote sensing of daily evapotranspiration and gross primary productivity of four forest ecosystems in East Asia using satellite multi-channel passive microwave measurements, Agric. For. Meteorol., 339, 109595. 10.1016/j.agrformet.2023.109595, 2023.

Xu, J., Liang, S., Ma, H. and He, T.: Generating 5 km resolution 1981–2018 daily global land surface longwave radiation products from AVHRR shortwave and longwave observations using densely connected convolutional neural networks, Remote Sens. Environ., 280, 113223. 10.1016/j.rse.2022.113223, 2022.

Zhang, Y., Kong, D., Gan, R., Chiew, F. H. S., McVicar, T. R., Zhang, Q. and Yang, Y.: Coupled estimation of 500 m and 8-day resolution global evapotranspiration and gross primary production in 2002–2017, Remote Sens. Environ., 222, 165-182. 10.1016/j.rse.2018.12.031, 2019.

Zheng, C., Jia, L. and Hu, G.: Global land surface evapotranspiration monitoring by ETMonitor model driven by multi-source satellite earth observations, J. Hydrol., 613, 128444. 10.1016/j.jhydrol.2022.128444, 2022.

---

## Author Comment (AC2)

**Responses to the Comments and Suggestions**

Reviewer #1:

This paper presents an energy conservation datasets of global land surface radiation and heat fluxes from 2000 to 2020. The dataset is generated by the model of Coordinated estimates of land Surface Energy Balance components (CoSEB), with a combination of GLASS and MODIS remote sensing data, ERA5-Land reanalysis datasets, topographic data, CO2 concentration data, and observations at 258 eddy covariance sites worldwide from the AmeriFlux, FLUXNET, EuroFlux, OzFlux, ChinaFLUX and TPDC. The primary merit of this new model is energy-conservation. Although the dataset might be useful, this dataset is not the first energy conservation datasets of global land surface radiation and heat fluxes as claimed by the authors. Therefore, major revisions are required before the paper is accepted.

Ans: Thank you very much for your valuable comments and suggestions. We sincerely appreciate your recognition of the dataset and the CoSEB model's merit in ensuring energy conservation. We would like to clarify that our initial statement, which described the datasets as "the first energy-conservation datasets of global land surface radiation and heat fluxes," may not have been entirely accurate. After careful consideration, we have revised the manuscript to more precisely describe the datasets as "the first data-driven energy-conservation datasets of global land-surface radiation and heat fluxes". Besides, we have carefully considered all the comments and suggestions from you and another reviewer and made corresponding modifications and clarifications in the revised manuscript. More detailed information of our revisions can be found in the item-by-item response below.

Specific comments:

1. The authors claim that "This study presents the first energy conservation datasets of global land surface radiation and heat fluxes", but reanalysis datasets, such as ERA5 which is used as inputs of this new dataset, also provide energy conservation surface fluxes for these energy fluxes. Maybe the authors want to say that this is the first remote sensing-based dataset? But the ERA5 radiative fluxes, which are not remote sensing-based, are used to generate surface fluxes in this paper, so this dataset is neither the first remote sensing-based dataset.

Ans: We sincerely thank the reviewer for this insightful comment. We acknowledge that reanalysis datasets, such as ERA5-Land, can in principle calculate these fluxes based on surface energy conservation. However, these reanalysis datasets rarely include all eight flux components directly. For example, ERA5-Land does not explicitly provide upward shortwave radiation, upward longwave radiation, net radiation or soil heat flux. Additionally, we would also like to clarify that the CoSEB-based datasets were developed by integrating both remote sensing products (e.g., PTC from MOD44B, LAI and FVC from GLASS, DEM, slope, and aspect from GMTED2010) and meteorological reanalysis data as inputs. It should be noted that widely used surface radiation and heat flux products, commonly referred to as remote sensing-based datasets, generally require meteorological reanalysis data as

inputs, e.g., the MOD16 ET product (Mu et al., 2011), SSEBop ET product (Senay et al., 2020), and GLASS radiation products (Wang et al., 2015; Xu et al., 2022), rather than relying solely on remote sensing data. Therefore, although our CoSEB-based datasets incorporate meteorological data from ERA5-Land in addition to remote sensing data, we believe it appropriate to refer to them as remote sensing-based datasets.

After careful consideration, we have revised the manuscript to more precisely describe the datasets as "the first data-driven energy-conservation datasets of global land-surface radiation and heat fluxes". We have revised this in the new manuscript as follows:

Abstract:

"This study presents the first data-driven energy-conservation datasets of global land surface radiation and heat fluxes from 2000 to 2020 ... The developed CoSEB-based datasets are strikingly advantageous in that [1] they are the first data-driven global datasets that satisfy both surface radiation balance ($SWI_N$ - $SW_{OUT}$ + $LW_{IN}$ - $LW_{OUT}$ = Rn) and heat balance (LE + H + G = Rn) among the eight fluxes,…"

5 Discussion

"The main advantages of our CoSEB-based datasets of land surface radiation and heat fluxes lie in that [1] they are the first data-driven global datasets that satisfy both surface radiation balance ($SWI_N$ - $SW_{OUT}$ + $LW_{IN}$ - $LW_{OUT}$ = Rn) and heat balance (LE + H + G = Rn) among the eight fluxes, as demonstrated by both the RIR and EIR of 0, …"

"Despite these uncertainties, it is worth emphasizing that our work was the first attempt to innovatively develop data-driven energy-conservation datasets of global land surface radiation and heat fluxes with high accuracies."

7 Summary and Conclusion

"This study for the first time developed data-driven energy-conservation datasets of global land surface radiation and heat fluxes…"

"The CoSEB-based datasets of land surface radiation and heat fluxes are the first data-driven global datasets that satisfy both surface radiation balance ($SWI_N$ - $SW_{OUT}$ + $LW_{IN}$ - $LW_{OUT}$ = Rn) and heat balance (LE + H + G = Rn) among the eight fluxes."

2. The merit of this new dataset is still unclear to me. According to Lines 171-180, ERA5 downward solar radiation and net thermal radiation at the surface is used in this paper, but why not simply use ERA5 fluxes if someone need to surface fluxes? The new dataset might be more accurate than ERA5 in places where ground-based observations are used to generate the new dataset, but the ground sites are sparce. To solve this problem, the authors should compare in-situ measurements with both the new data and ERA5 data in independent sites (i. e., sites that are not used in the generation of the new dataset).

Ans: We sincerely appreciate the reviewer's insightful comment and suggestion. We would like to clarify that the ERA5-Land reanalysis datasets do not explicitly provide upward shortwave radiation, upward longwave radiation, net radiation, or soil heat flux, although these components can theoretically be computed using surface radiation and heat balance principles. The purpose of our work was to innovatively provide energy-conservation surface radiation and heat fluxes based on data-driven technique. This is motivated by the fact that existing data-driven products (e.g., FLUXCOM and GLASS) estimate each energy component separately, leading to obvious energy imbalance among these components (Wang et al., 2025).

To further address the reviewer's concern, we have compared estimates from CoSEB-based datasets and ERA5-Land datasets with in-situ observations from 44 sites (collected from recently published JapanFlux and updated AmeriFlux, see the sites for "test" in Table S1), which are independent from the 258 sites that are used for model construction and datasets generation. As demonstrated by the comparison results (see Figs. S6 and S7), the CoSEB-based datasets exhibit higher accuracy than the ERA5-Land datasets in estimating surface energy fluxes, especially in estimating $SW_{OUT}$, H and G. We have discussed this in the third paragraph of Section 5 in the revised manuscript with the following sentences:

"Furthermore, the CoSEB-based datasets outperformed the ERA5-Land reanalysis datasets in estimating surface energy fluxes (where $SW_{OUT}$, $LW_{OUT}$, Rn and G for the ERA-Land were inferred from surface radiation balance and heat balance), particularly for $SW_{OUT}$, H and G, with RMSE reductions of 0.13-8.15 W/m$^2$ when validated against in situ observations at the 44 test sites (Figs. S6 and S7 in the Supplementary Material)."

[Figure]

**Fig. S6 Comparison of the daily downward shortwave radiation ($SW_{IN}$), upward shortwave radiation ($SW_{OUT}$), downward longwave radiation ($LW_{IN}$), upward longwave radiation ($LW_{OUT}$) and net radiation (Rn) from the CoSEB-based datasets (upper 5 panels) and ERA5-Land (lower 5 panels) with the in-situ observed $SW_{IN}$, $SW_{OUT}$, $LW_{IN}$ and $LW_{OUT}$ at 44 test sites. The colorbar represents the normalized density of data points.**

[Figure]

**Fig. S7 Comparison of the daily latent heat flux (LE), sensible heat flux (H) and soil heat flux (G) from the CoSEB-based datasets (first row) and ERA5-Land (second row) with the in-situ energy imbalance-corrected LE ( $LE_{daily}^{corr}$ ) and H ( $H_{daily}^{corr}$ ), as well as observed G at 44 test sites. The colorbar represents the normalized density of data points.**

3. The abstract is not well formatted. An abstract usually provides a brief and comprehensive summary, so trivial details in brackets [including downward shortwave radiation (SWIN), downward longwave radiation (LWIN), upward shortwave 15 radiation (SWOUT), upward longwave radiation (LWOUT) and net radiation (Rn)], [including latent heat flux (LE), soil heat flux (G) and sensible heat flux (H)], and (SWIN - SWOUT + LWIN - LWOUT = Rn) might be deleted. Internet links https://doi.org/10.11888/Terre.tpdc.302559 and citations (Tang et al., 2025a) should be removed from the abstract. On the other hand, the authors should briefly describe how these data sources are used to generate the new dataset.

Ans: We appreciate the reviewer's suggestion. We would like to clarify that the latter part of the Abstract describes the accuracy of each of the eight surface radiation and heat flux components, as well as the overall surface radiation balance and energy balance among them. Therefore, to ensure consistency and readability, we chose to retain the introduction of all eight fluxes and their corresponding abbreviations at the beginning of the Abstract. However, the two equations, ($SW_{IN}$ - $SW_{OUT}$ + $LW_{IN}$ - $LW_{OUT}$ = Rn) and (LE + H + G = Rn), were deleted in the Abstract, as suggested by the reviewer. Furthermore, the links and citations of the datasets are mandatorily required by the journal and editors in the Abstract, and therefore cannot be removed. Besides, following the reviewer's suggestion, we have briefly explained how multiple data sources were integrated to generate the CoSEB-based datasets in the revised manuscript as follows:

"This study presents the first data-driven energy-conservation datasets of global land surface radiation and heat fluxes from 2000 to 2020, generated by our model of Coordinated estimates of land Surface Energy Balance components (CoSEB). The model integrates GLASS and MODIS remote sensing data, ERA5-Land reanalysis datasets, topographic data, $CO_2$ concentration data as independent variables and in situ radiation and heat flux observations at 258 eddy covariance sites worldwide as dependent variables within a multivariate random forest technique to effectively learn the physics of energy conservation."

Reference:

Mu, Q., Zhao, M. and Running, S. W.: Improvements to a MODIS global terrestrial evapotranspiration algorithm, Remote Sens. Environ., 115, 1781-1800. 10.1016/j.rse.2011.02.019, 2011.

Senay, G. B., Kagone, S. and Velpuri, N. M.: Operational Global Actual Evapotranspiration: Development, Evaluation and Dissemination, Sensors (Basel), 20. 10.3390/s20071915, 2020.

Wang, D., Liang, S., He, T. and Shi, Q.: Estimation of Daily Surface Shortwave Net Radiation From the Combined MODIS Data, IEEE Trans. Geosci. Remote Sensing, 53, 5519-5529. 10.1109/tgrs.2015.2424716, 2015.

Wang, J., Tang, R., Liu, M., Jiang, Y., Huang, L. and Li, Z.-L.: Coordinated estimates of 4-day 500 m global land surface energy balance components, Remote Sens. Environ., 326, 114795. 10.1016/j.rse.2025.114795, 2025.

Xu, J., Liang, S., Ma, H. and He, T.: Generating 5 km resolution 1981–2018 daily global land surface longwave radiation products from AVHRR shortwave and longwave observations using densely connected convolutional neural networks, Remote Sens. Environ., 280, 113223. 10.1016/j.rse.2022.113223, 2022.

---

## Editor Decision (ED1)

Review of ESSD

Authors describe use of 'coordinated' model (CoSEB) to understand surface energy and heat budgets. Their paper presents reasons that (many?) others might do likewise.

Two reviewers offered good thoughtful comments, including substantive suggestions and concerns. Authors responded in detail, including by revising perview and re-running some calculations. This reviewer considers that authors have made useful appropriate response. Authors now present a well-written well-organized manuscript. Data prove easy to access and to use.

I raise general questions, focused more on impact and utility. Topical editor will need to decide based on these comments

Their product covers, unfortunately, northern hemisphere land. Admittedly we can't escape land only. But, without Australia, we would basically have no data from southern hemisphere. Not in any way the fault of these authors, but they do need to admit up front these limitations. Do we miss significant ecosystems or soil types from South America or South Africa?

This reader remains confused about how CoSEB assures surface energy or surface heat budgets close to 'zero'. Authors state this closure multiple times as a positive feature. I suppose multiple internal calculations and iterations must assure this result, and I applaud their 'clean' outcome, but what artificial corrections will we have introduced or accepted to ensure this outcome? I doubt that we know (can measure) LE or RN to that accuracy? But the model can do it? What happens seasonally, as ecosystems burn or migrate, or as one feature or multiple features change? CoSEB always ensures zero net offset? I appreciate what we might gain. What might we lose?

The standard deviations in time series (Fig 14) give me pause. All high! Very high. Hard to tell exactly, but CoSEB seems no better than any individual or cumulative product. So what have we lost or gained? If no product shows significant trends, why have we made such effort? If no product show significant change with time, over 20 years, we do not track relevant features? We all know that global atmospheric $CO_2$ rises significantly over that time. Likewise atmospheric temperature. But none of these 'measured' fluxes? Land static, everything driven by ocean?

I likewise doubt differences between coordinated (CoSEB) and uncoordinated analyses. In Figs 3 vs 4, considering r2, CoSEB shows no significant differences to uncoordinated RF analyses. If true, why do authors and readers need to spend additional time comparing validations or imaging geographic differences?

I emphasize: I like coordinated integrated approach! I respect authors skil in pulling this together. I worry - and authors can correct - that we might oversell.

---

## Author Response (AR2)

**Responses to the Comments and Suggestions**

Dear Ronglin Tang,

We are pleased to inform you that the topic editor report for the following ESSD manuscript is now available:

essd-2025-456

Title: Energy-conservation datasets of global land surface radiation and heat fluxes from 2000-2020 generated by CoSEB

Author(s): Junrui Wang et al.

MS type: Data description paper

Iteration: Minor revision

The topic editor has decided that minor revisions are necessary before the manuscript can be accepted. Please log in using your Copernicus Office user ID  to find the topic editor report at: https://editor.copernicus.org/ESSD/ms_records/essd-2025-456

We kindly ask you to revise your manuscript accordingly and to upload the revised files, a point-by-point reply to the comments, and a marked-up manuscript version showing the changes made no later than 21 Dec 2025 at: https://editor.copernicus.org/ESSD/review-file-upload/essd-2025-456

Please find all information on manuscript submission at: https://www.earth-system-science-data.net/for_authors/submit_your_manuscript.html

Your revised manuscript will be reviewed again and you will be informed about the outcome by separate email.

Besides adjustments requested by the topic editor or referees, please check your manuscript carefully for typos, missing co-authors and their affiliations, terminology, updates of data in tables, or updates of variables in equations. All these have to be clarified with the topic editor and therefore have to be included before you submit your revised manuscript. Should your manuscript be finally accepted it will not be possible to include such rather substantial changes anymore when your manuscript is in final production (proofreading).

Please note that all referee and editor reports, the author's response, as well as the different manuscript versions of the peer-review completion (post-discussion review of revised submission) will be published if your paper will be accepted for final publication in ESSD.

You are invited to monitor the processing of your manuscript via your MS overview at: https://editor.copernicus.org/ESSD/my_manuscript_overview

In case any questions arise, please do not hesitate to contact me. Thank you very much for your cooperation.

Kind regards,

The editorial support team

Copernicus Publications editorial@copernicus.org

Thank you very much for your excellent revision work. We appreciate your thorough responses to the reviewers' comments.

After integrating additional feedback from further reviewers, we encourage authors to revise the manuscript to accommodate their suggestions.

Reviewer 1 wrote:

"Authors have done an excellent job with their responses. I am happy to approve the publication.

I did want to follow up on one point. While it's true, as the authors point out, that Earth system models have large ensemble runs at 1 degree, currently several Earth system models are being deployed at finer scales (0.5 degrees, even 1 km). In general, I would like the authors to mention that their framework can reproduce results of Earth system models. This would be a great benefit to the community given the effort required to run those models."

In addition, Reviewer 2's suggestions are provided in the attached file. I agree with Reviewer 2 that the manuscript should explicitly address the limitations of the current dataset, particularly the lack of data coverage in the Southern Hemisphere, in order to avoid potentially misleading users.

We would be grateful if you could revise the manuscript to address these comments and resubmit a revised version for further consideration.

Additional private note (visible to authors and reviewers only):

Thank you very much for your support!

Ans: We sincerely thank the editor and reviewers for their time and effort, as well as for their valuable comments and constructive suggestions, which have greatly helped to improve the quality of our manuscript.

In response to the suggestion from Reviewer 1, we have further clarified the capability of CoSEB to provide significant benefits to the community, particularly in light of the high computational cost and long execution times associated with Earth system models. This clarification has been added at the end of the third paragraph of Section 5 with the following sentences:

"Preliminary analysis indicates that the CoSEB-based datasets exhibit spatial patterns consistent with those of mainstream RS-based datasets and Earth system model outputs (see Fig. S9 in the supplementary material), suggesting that the CoSEB-based datasets (or CoSEB framework) more broadly, are capable of reproducing the large-scale spatial features of Earth system models. This capability would be a great benefit to the community given the limitations associated with the high computational cost and long execution time of Earth system models. More detailed analysis about their similarities and differences can be further conducted in future work."

Regarding the suggestion from Reviewer 2, we have explicitly acknowledged the limitation arising from the uneven distribution of eddy-covariance sites used in this study and emphasized the need for additional observations from currently underrepresented regions of the Southern Hemisphere to improve the global representativeness of the CoSEB-based datasets in the future in the last paragraph of

Section 5 in the revised manuscript with the following sentences:

"Note that the 302 sites used for training, validation, and testing are predominantly located in the Northern Hemisphere, reflecting the inherent uneven distribution of the global flux networks. Although these sites cover a wide range of land cover types and climate regimes, thereby providing substantial heterogeneity for model development, the limited representation of the Southern Hemisphere may introduce uncertainties in the estimation of surface radiation and heat fluxes for certain ecosystems and soil types. In the future, enhancing the flux observation network coverage in the Southern Hemisphere, particularly in South America and Africa, and incorporating these observations into the CoSEB framework would help further improve the accuracy of surface radiation and heat flux estimates in these regions."

More detailed information on our revisions to other comments and suggestions can be found in the item-by-item response below.

Reviewer #1:

Authors have done an excellent job with their responses. I am happy to approve the publication.

I did want to follow up on one point. While it's true, as the authors point out, that Earth system models have large ensemble runs at 1 degree, currently several Earth system models are being deployed at finer scales (0.5 degrees, even 1 km). In general, I would like the authors to mention that their framework can reproduce results of Earth system models. This would be a great benefit to the community given the effort required to run those models.

Ans: Thanks for your valuable comments and constructive suggestions. Following your suggestion, we have further clarified this at the end of the third paragraph of Section 5 in the revised manuscript with the following sentences:

"Preliminary analysis indicates that the CoSEB-based datasets exhibit spatial patterns consistent with those of mainstream RS-based datasets and Earth system model outputs (see Fig. S9 in the supplementary material), suggesting that the CoSEB-based datasets (or CoSEB framework) more broadly, are capable of reproducing the large-scale spatial features of Earth system models. This capability would be a great benefit to the community given the limitations associated with the high computational cost and long execution time of Earth system models. More detailed analysis about their similarities and differences can be further conducted in future work."

Reviewer #2:

Authors describe use of 'coordinated' model (CoSEB) to understand surface energy and heat budgets. Their paper presents reasons that (many?) others might do likewise.

Ans: The lack of energy closure in existing products/datasets represents a fundamental and well-recognized issue, posing substantial challenges for studies of surface energy balance changes under a changing climate. We believe that this concern is shared by many others in the community. The key contribution of our study is to address this issue by developing the CoSEB framework, which enables coordinated multi-component estimation while explicitly enforcing surface energy conservation.

Two reviewers offered good thoughtful comments, including substantive suggestions and concerns. Authors responded in detail, including by revising perview and re-running some calculations. This reviewer considers that authors have made useful appropriate response. Authors now present a well-written well-organized manuscript. Data prove easy to access and to use.

I raise general questions, focused more on impact and utility. Topical editor will need to decide based on these comments

Ans: We sincerely thank the reviewer for the thoughtful and encouraging assessment of our revised manuscript. Regarding the reviewer's general questions on impact and utility, we have carefully considered these points and further revised the manuscript to better articulate the scientific value, applicability, and potential for wider adoption of our CoSEB-based datasets. We believe our manuscript has been greatly improved by following the reviewer's comments and suggestions.

Their product covers, unfortunately, northern hemisphere land. Admittedly we can't escape land only. But, without Australia, we would basically have no data from southern hemisphere. Not in any way the fault of these authors, but they do need to admit up front these limitations. Do we miss significant ecosystems or soil types from South America or South Africa?

Ans: Thanks for your valuable comment and question. We would like to clarify that our CoSEB-based datasets provide global land estimates, including those of the Southern Hemisphere. We understand that this concern likely arises from the fact that the eddy-covariance sites used in this study are predominantly located in the Northern Hemisphere. We acknowledge that, despite our efforts to collect observations from multiple global/regional networks (e.g., FLUXNET, AmeriFlux, EuroFlux, OzFlux, JapanFlux, and TPDC), the 302 sites retained after data preprocessing and quality control are mainly concentrated in the Northern Hemisphere. This limitation is inherent to the current global eddy-covariance observational network and is not specific to our study. We agree that the sparse coverage of the Southern Hemisphere, particularly South America and South Africa, may limit the representation of some ecosystems and soil types. Nevertheless, the selected sites encompass 14 land cover types (e.g., evergreen needleleaf forests, evergreen broadleaf forests, savannas, croplands, and grasslands) across a wide range of climate regimes (tropical, dry, temperate, and continental). The multi-year observations capture diverse meteorological, soil moisture, and vegetation conditions, providing substantial spatiotemporal variability for model development.

Responding to the reviewer's comments, we have explicitly acknowledged this limitation and emphasized the need for additional observations from currently underrepresented Southern Hemisphere regions to improve the global representativeness of the CoSEB-based datasets in the future in the last paragraph of Section 5 in the revised manuscript with the following sentences:

"Note that the 302 sites used for training, validation, and testing are predominantly located in the Northern Hemisphere, reflecting the inherent uneven distribution of the global flux networks. Although these sites cover a wide range of land cover types and climate regimes, thereby providing substantial heterogeneity for model development, the limited representation of the Southern Hemisphere may introduce uncertainties in the estimation of surface radiation and heat fluxes for certain ecosystems and soil types. In the future, enhancing the flux observation network coverage in the Southern Hemisphere, particularly in South America and Africa, and incorporating these observations into the CoSEB framework would help further improve the accuracy of surface radiation and heat flux estimates in these regions."

This reader remains confused about how CoSEB assures surface energy or surface heat budgets close to 'zero'. Authors state this closure multiple times as a positive feature. I suppose multiple internal calculations and iterations must assure this result, and I applaud their 'clean' outcome, but what artificial corrections will we have introduced or accepted to ensure this outcome? I doubt that we know (can measure) LE or RN to that accuracy? But the model can do it? What happens seasonally, as ecosystems burn or migrate, or as one feature or multiple features change? CoSEB always ensures zero net offset? I appreciate what we might gain. What might we lose?

Ans: Thank you for your insightful comments and questions. We would like to clarify that the radiation and heat balance in this study refers specifically to the conservation among the eight variables analyzed in the manuscript (i.e., $SW_{IN}$, $LW_{IN}$, $SW_{OUT}$, $LW_{OUT}$, Rn, LE, G, H), which constitute the major components of the surface energy budget, and does not account for energy introduced by disturbance-related processes such as wildfires and volcanic eruptions. We have further illustrated this in the last paragraph of Section 5 in the revised manuscript with the following sentences:

"Furthermore, the radiation and heat balance in this study refers specifically to the conservation among the eight variables (i.e., $SW_{IN}$, $LW_{IN}$, $SW_{OUT}$, $LW_{OUT}$, Rn, LE, G, H), which constitute the major components of the surface energy budget, and does not account for energy introduced by disturbance-related processes such as wildfires and volcanic eruptions."

The standard deviations in time series (Fig 14) give me pause. All high! Very high. Hard to tell exactly, but CoSEB seems no better than any individual or cumulative product. So what have we lost or gained? If no product shows significant trends, why have we made such effort? If no product show significant change with time, over 20 years, we do not track relevant features? We all know that global atmospheric CO2 rises significantly over that time. Likewise atmospheric temperature. But none of these 'measured' fluxes? Land static, everything driven by ocean?

Ans: We thank the reviewer for the thoughtful comments and questions. We would like to clarify that the standard deviation shown in Fig. 14 primarily reflects the spatial variability of each product across global land surfaces. Given the strong heterogeneity in climate regimes, land cover types, and surface energy partitioning, substantial spatial variability is expected and does not imply inferior product performance. Moreover, the performance of the CoSEB-based datasets was evaluated through independent validation and inter-comparison against in situ observations, with results from 44 test sites (see Section 4.2) consistently showing lower RMSE and bias, and higher $R^2$, than those of other mainstream products.

In addition, the relatively large mean values and standard deviations in Fig. 14 make it difficult to visually identify long-term trends in the time series. To better illustrate these trends, we have additionally added a figure (Fig. S4 in the Supplementary Material) presenting anomaly time series for each product. This anomaly-based representation reveals clear temporal trends over the 20 years. We have illustrated this in the third paragraph of Section 4.3 in the revised manuscript with the following sentences:

"The anomaly-based analyses (Fig. S4 in the Supplementary Material) reveal clear and coherent temporal trends of these radiation and heat fluxes, which respond well to global climate change, such as increasing atmospheric $CO_2$ and rising air temperatures."

[Figure]

**Fig. S4 Anomalies of downward shortwave radiation ($SW_{IN}$), upward shortwave radiation ($SW_{OUT}$), downward longwave radiation ($LW_{IN}$), upward longwave radiation ($LW_{OUT}$), net radiation (Rn), latent heat flux (LE), sensible heat flux (H) and soil heat flux (G) from 2001 to 2018 from the CoSEB-based datasets, BESS-Rad, GLASS, FLUXCOM, BESSV2.0, PML_V2, MOD16A2 and ETMonitor.**

I likewise doubt differences between coordinated (CoSEB) and uncoordinated analyses. In Figs3 vs 4, considering r2, CoSEB shows no significant differences to uncoordinated RF analyses. If true, why do authors and readers need to spend additional time comparing validations or imaging geographic differences?

Ans: Thank you for your questions. We would like to clarify that the key advantage of our coordinated model (CoSEB) over the uncoordinated models lies in its ability to enforce energy balance among the eight radiation and heat fluxes, thereby producing estimates with improved physical consistency. We have already illustrated this in the second paragraph of Section 4.1.1 and in the first paragraph of Section 4.1.2 with the following sentences:

"Strikingly, the CoSEB model exhibited large superiority in balancing the surface radiation and heat fluxes, with the radiation imbalance ratio [RIR, defined as $100\% \times (SW_{IN} - SW_{OUT} + LW_{IN} - LW_{OUT})/Rn$] and energy imbalance ratio [EIR, defined as $100\% \times (Rn - G - LE - H)/Rn$] of 0, while the RF-based uncoordinated models showed substantial imbalances of the surface radiation and heat fluxes, with RIR and EIR that were approximately normally distributed, having absolute mean values of 38.84% and 31.22%, respectively, and reaching as high as 50% in some cases."

"Furthermore, the CoSEB model demonstrated a large superiority in maintaining surface radiation balance among the five radiation components, with the RIR of 0, in contrast to the RF-based models, which failed to meet this balance, exhibiting significant RIR exceeding 50%."

In addition, the validation and inter-comparisons conducted at 44 independent test sites in Section 4.2 indicate that the CoSEB-based datasets consistently outperform existing mainstream products/datasets, as evidenced by lower RMSE and bias and higher $R^2$. In light of these advantages, we further conducted a comparison of the geographic patterns between the CoSEB-based datasets and existing products.

I emphasize: I like coordinated integrated approach! I respect authors skil in pulling this together. I worry - and authors can correct - that we might oversell.
Ans: We sincerely thank the reviewer for the positive comments and for recognizing the value of our coordinated approach. We have revised our manuscript following your comments and suggestions. We believe that, after this revision, the contributions of our work are no longer overstated.

[revised manuscript text omitted]